# A sex-specific thermogenic neurocircuit induced by predator smell recruiting cholecystokinin neurons in the dorsomedial hypothalamus

Predrag Jovanovic [1,2], Allan-Hermann Pool[3], Nancy Morones[1,2], Yidan Wang[1,2], Edward Novinbakht[1,2], Nareg Keshishian[1,2], Kaitlyn Jang[1,2], Yuki Oka [4] & Celine E. Riera [1,2,5]

Olfactory cues are vital for prey animals like rodents to perceive and evade predators. Stress-induced hyperthermia, via brown adipose tissue (BAT) thermogenesis, boosts physical performance and facilitates escape. However, many aspects of this response, including thermogenic control and sex-specific effects, remain enigmatic. Our study unveils that the predator odor tri-methylthiazoline (TMT) elicits BAT thermogenesis, suppresses feeding, and drives glucocorticoid release in female mice. Chemogenetic stimulation of olfactory bulb (OB) mitral cells recapitulates the thermogenic output of this response and associated stress hormone corticosterone release in female mice. Neuronal projections from OB to medial amygdala (MeA) and dorsomedial hypothalamus (DMH) exhibit female-specific cFos activity toward odors. Cell sorting and single-cell RNA-sequencing of DMH identify cholecystokinin (CCK)-expressing neurons as recipients of predator odor cues. Chemogenetic manipulation and neuronal silencing of DMH[CCK] neurons further implicate these neurons in the propagation of predator odor-associated thermogenesis and food intake suppression, highlighting their role in female stress-induced hyperthermia.

The detection of stressful cues initiates a complex combination of neuronal, behavioral and endocrine responses which ensure organismal survival. It is well established that neural circuits in the hypothalamus are essential for coordinating mammalian responses to perceived threats. A fundamental hypothalamic neuronal population involved in promoting endocrine responses to stress is comprised of corticotropin-releasing hormone (CRH) neurons found in the paraventricular nucleus (PVH)[1,2]. Their stimulation controls the release

of cortisol in humans, and its rodent analog corticosterone (CORT) through hypothalamic-pituitary-adrenal (HPA) axis activity, initiated by release of CRH from the PVH into the pituitary portal[3]. CRH neurons have been shown to be important in mediating other stress-related functions, including changes in behavior[2] and encoding of valence[4].

A key role of the stress response is the mobilization of the body's energy stores to promote an adaptive response, such as fleeing a dangerous situation, however acute and chronic activation of stress

[1]Center for Neural Science and Medicine, Department of Biomedical Sciences, Cedars-Sinai Medical Center, 127 South San Vicente Boulevard, Los Angeles, CA 90048, USA. [2]Board of Governors Regenerative Medicine Institute, Cedars-Sinai Medical Center, 127 South San Vicente Boulevard, Los Angeles, CA 90048, USA. [3]Department of Neuroscience, Department of Anesthesiology and Pain Management, Peter O'Donnell Jr. Brain Institute, University of Texas Southwestern Medical Center, Dallas, TX, USA. [4]Division of Biology and Biological Engineering, California Institute of Technology, Pasadena, CA, USA. [5]Department of Neurology, Cedars-Sinai Medical Center, 127 South San Vicente Boulevard, Los Angeles, CA 90048, USA. ✉e-mail: Celine.riera@cshs.org

share different outcomes on energy metabolism[5]. Yet, the current understanding on the neurocircuits associated with the different metabolic consequences of acute and chronic stress is limited. Acute CRH release mobilizes energy stores, including increasing brown adipose tissue (BAT) thermogenesis, enhancing locomotion and suppressing feeding to ensure a rapid response favoring animal survival, in a process recognized as stress-induced hyperthermia[6–8]. In contrast, repeated stress is associated with HPA overactivity and chronic elevation of CORT[5]. Prolonged elevation of CORT levels in rodents promotes weight gain and higher adiposity, through the modulation of feeding behavior leading to hyperphagia[9], without impacting energy expenditure[10–12].

Although stress-induced hyperthermia is a fundamental autonomic stress response observed in many mammalian species, the central neuronal players required for this response have yet to be characterized. Stress-induced hyperthermia has been observed in the context of psychological stress, under the perception of danger, discomfort or pain and upon central delivery of CRH[8,13,14]. These reports have pointed at the dorsomedial hypothalamus (DMH) as a crucial recipient and mediator of the stress hyperthermic response. In addition, perception of the fear-inducing odor 2,4,5-trimethylthiazole (mT), a synthetic compound related to the red fox predator scent 2,4,5-trimethylthiazoline (TMT), is associated with cFos upregulation in the DMH and an elevation of plasma CORT in mice, compared to control scent[15]. Other predator odors commonly used in laboratory settings, including cat and ferret odors, have been linked to the stimulation of ventromedial hypothalamus (VMH) neurons for the propagation of fear and aggressive responses[16–18]. The representation of diverse predator odors is highly organized and spatially localized to subregions of the medial amygdala (MeA) and VMH[17,18]. The MeA has been implicated in fear and anxiety responses[19], whereas the VMH has been associated with defensive, escape, avoidance and panic-like behaviors[16–18,20,21]. Among the various predator odors commonly used in lab settings driving avoidance behavior in mice, TMT specifically led to stimulus burying in the cage, and did not induce VMH neurons stimulation[18], suggesting that this stressful odor is associated with a specific avoidance behavior possibly mediated by distinct neuronal targets.

It is well appreciated that the stress response is sexually dimorphic, with women having a higher incidence of anxiety disorders including panic disorders, and trauma-related disorders such as post-traumatic stress disorders[22]. In rodents, presynaptic innervation using synaptophysin staining density in female rats has revealed higher innervation preceding chronic variable stress in regions involved in stress processing, including paraventricular nucleus of hypothalamus (PVH), prefrontal cortex (PFC), bed nucleus of the stria terminalis (BST) and basolateral amygdala[23]. However, after 14 days of chronic variable stress, PVH, BST and amygdala had reduced synaptophysin density in females but not in males. These data suggest that chronic stress may be associated with increased susceptibility to stress-associated disorders in females, yet the neurons involved in sexually dimorphic regulation of stress are not known. In the present study, we sought to identify neuronal populations involved in stress-induced hyperthermia. Using a combination of predator odor presentation, as well as chemogenetic manipulation of olfactory neurons, we found the existence of a sexually dimorphic regulation of thermogenesis which occurs specifically in females. Combined anterograde tracing and single cell RNA-sequencing identify DMH$^{CCK}$ neurons as a recipient of olfactory signals and modulators of feeding and energy expenditure in the female rodent brain.

## Results

### Fear odor detection promotes a sexually dimorphic regulation of energy expenditure
Because of a pre-existing link between the predator smell TMT and weight loss in diet-induced obesity animals[15], we explored the effects of

this aversive odor on energy homeostasis. Naïve and normal chow fed C57BL/6 mice were exposed to TMT or control scent using a cotton swab while subjected to indirect calorimetry with the swab present throughout the experiment (Fig. 1a). Remarkably, CORT levels remained elevated in TMT-exposed females compared to males after 90 min (Fig. 1b). Swab introduction with control odor led to a transient increase in oxygen consumption (VO$_2$) in both sexes, combined with higher activity for a time period of 45–60 min (Fig. 1c–f). In males, VO$_2$ and activity were similar among control and TMT-treated mice for the first 2 h after swab introduction (Fig. 1c, e). In females, subsequently upon predator odor introduction, a significantly higher VO$_2$ response to TMT was observed, together with elevated locomotor activity which persisted for 60–90 min (Fig. 1d, f). Female's gain of VO$_2$ was observed independently of estrus cycle phase (Supplementary Fig. 1a, b). Food intake was negligible in the 3 h following odor detection in ad libitum-fed male and female cohorts, with no significant differences between TMT and control animals (Fig. 1g, h). Using infrared imaging, we measured the temperature of the brown adipose tissue (BAT) region upon odorant presentation, as BAT plays an essential role in adaptive thermogenesis and energy expenditure[24,25]. TMT odor was associated with a female-specific BAT temperature increase, peaking 15 min after odor introduction and coinciding with the maximum VO$_2$ effect (Fig. 1i, j). To further investigate the impact of TMT on feeding, we assessed refeeding after an overnight fast in the continuous presence of an odorant stimulus (TMT or control scent). Food intake was reduced in males and females exposed to TMT, with a more pronounced effect in females, where exposure to predator odor inhibited feeding (Fig. 1k, l). These results indicate that TMT exposure drives a sex-specific stress response, characterized by increased stress hormone production, higher movement and energy expenditure in female mice upon predator odor detection.

We then probed the sexually dimorphic stimulation of odor presentation within the mediobasal hypothalamus (MBH) in light of the recognized contribution of this region to the integration of food and predator cues[16,17,26,27]. Naïve mice of either sex were exposed to either a food odor employing peanut butter oil scent, TMT as a predator odor, and opposite sex urine scent as a pheromone scent (Fig. 1m). We validated these odors' ability to stimulate olfactory processing center regions including the main olfactory bulb (MOB), accessory olfactory bulb (AOB) and MeA nuclei (Supplementary Fig. 1c, d). As expected[3,4], volatile opposite sex urine exposure drove cFos neuronal activity in AOB (Supp Fig. 1c). The posterior medial amygdala (pMeA), mainly responsible for modulating responses related to reproduction[5,6] and cues of predator presence[7], was activated by opposite-sex odor and TMT exposure in both sexes (Supplementary Fig. 1d). TMT odor elevated neuronal activity within the ARC of both sexes (Fig. 1n, o). In line with previous observations[15], TMT exposure led to a high cFos induction in the DMH in males, but a more pronounced cFos response was visible in the female DMH (Fig. 1p, q). Exposure to different odors did not change neural activity in VMH and lateral hypothalamus (LH) (Supplementary Fig. 1e, f). Interestingly, while all animals were calorie replete during the experiments, peanut butter smell increased ARC cFos activity selectively in females (Fig. 1n, o).

### Chemogenetic stimulation of mitral and tufted olfactory cells increases energy expenditure in females
Our results indicate that a select aversive odor can drive changes in female energy expenditure, feeding and glucocorticoid release. The detection of odors in the nasal cavity is mediated by olfactory sensory neurons (OSNs) which send axons to a specific projection site in the olfactory bulb (OB). These axons form a glomerular structure where each glomerulus integrates afferent signals from thousands of OSNs. The output is then relayed via several dozen mitral/tufted cells[28]. Mitral/tufted projections are distributed via the lateral olfactory tract to a heterogeneous assemblage of secondary structures collectively labeled as the olfactory cortex. The piriform cortex and the cortical

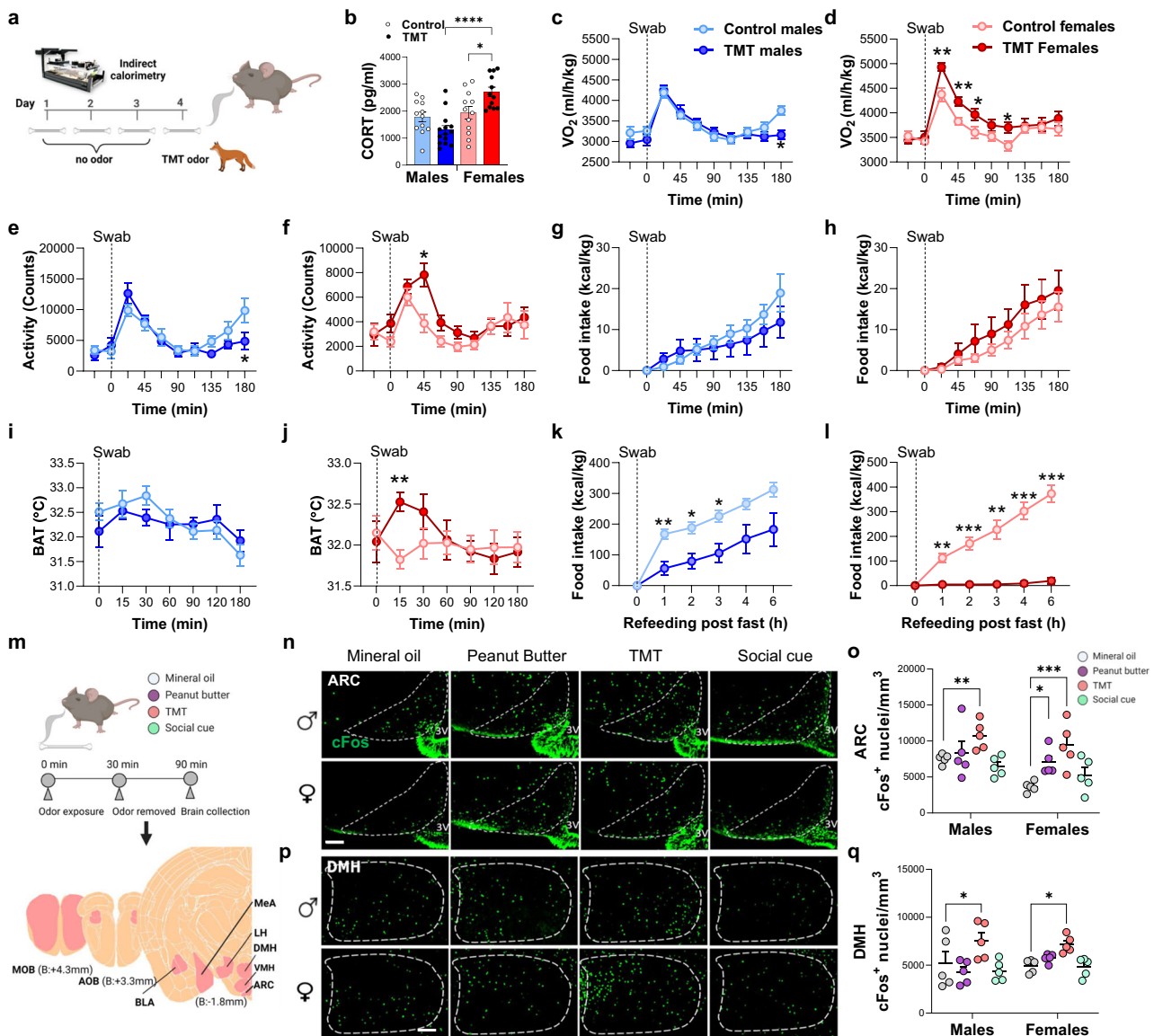

**Fig. 1 | Effect of predator smell on energy metabolism and mediobasal hypothalamus activity. a** Experimental design. Mice given TMT odor or control swab were assessed by indirect calorimetry. **b** CORT levels measured 90 min following TMT or control odor stimulus, males (light blue/blue) and females (pink/red),-males, $N=12$/control and $N=14$/TMT, females, $N=12$ per group, Two-way Anova with Tukey's post hoc comparison. **c, d** VO$_2$ post-exposure to TMT, $N=16$ per group, Two-way Anova with Sidak's post hoc comparison. **e, f** Activity 3 h post-exposure to TMT, males, $N=16$ per group, Two-way Anova with Sidak's post hoc comparison. **g, h** Food intake 3 h post-exposure to TMT, males, $N=11$ per group, females, $N=12$/control and $N=10$/TMT. **i, j** Brown adipose tissue temperature 3 h post-exposure to TMT, males, $N=11$ per group, females, $N=11$ per group, Two-way Anova with Sidak's post hoc comparison. **k, l** Food intake after an overnight fast in male and female mice exposed to TMT or control scent, males, $N=9$ control group and $N=10$ TMT group females, $N=10$ per group, Two-way Anova with Sidak's post hoc comparison. **m** Schematic of the experiment. Mice given an odor were monitored for changes in brain cFos expression. **n, o** cFos expression after odor exposure in ARC and quantification, $N=5$ per group, Two-way Anova with Dunnett's post hoc comparison. **p, q** cFos expression after odor exposure in DMH and quantification, $N=5$ per group, Two-way Anova with Dunnett's post hoc comparison. All bar graphs are presented as mean values ± SEM. Table S1 contains the detailed results of the statistical analysis. *$p < 0.05$, **$p < 0.01$, ***$p < 0.001$. Source data are provided as a Source Data file. corticosterone (CORT), arcuate nucleus of the hypothalamus (ARC), dorsomedial hypothalamus (DMH), 2,4,5-trimethylthiazoline (TMT). Scale bar 100 μm.

amygdala are the main recipients of inputs from the main OB (MOB), and transmit olfactory information to the orbitofrontal cortex, the insular cortex and hypothalamus. In order to gain a deeper understanding of the link between olfactory circuits and energy balance, we assessed the output of MOB mitral/tufted cell stimulation in the sexually dimorphic regulation of stress and energy balance. Given the large size of the MOB, we employed a genetic approach to express an activatory designer receptor exclusively activated by designer drug (DREADD)[29] within all mitral/tufted cells of the MOB (Fig. 2a). We generated Tbx21-Cre;hM3D$^{loxP/loxP}$ transgenic mice expressing the activatory Gq DREADD human M3 muscarinic acetylcholine receptors

(hM3D) in mitral and tufted cells of the MOB to test behavioral and metabolic phenotypes resulting from OB$^{Tbx21}$ stimulation. We verified the restriction of Cre expression within the MOB[30] (Supplementary Fig. 2a). Injection of the DREADD agonist clozapine-N-oxide (CNO) promoted cFos expression in mitral cells labelled by mCitrine, which marks hM3D$^+$ neurons (Fig. 2b, c). Female Cre$^+$ mice showed increased ability to locate a buried food pellet, indicating higher motivation, which was not observed in males (Fig. 2d). Anxiety-like behavior measured using the percent time spent in the center of an open field was also highly increased in female Cre$^+$ mice over female Cre$^-$ controls, as observed by reduced time spent in the central area, whereas

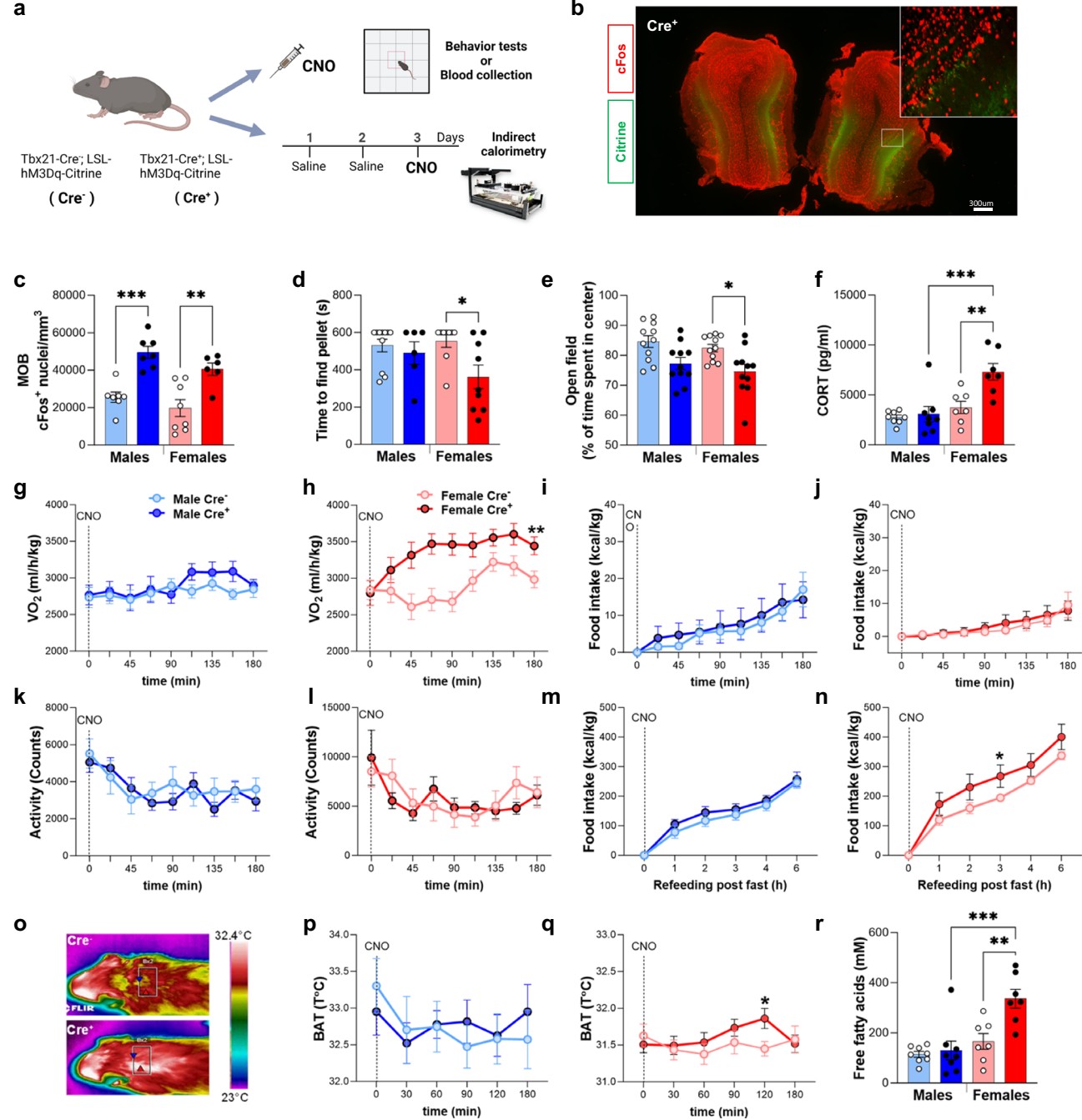

**Fig. 2 | Chemogenetic stimulation of olfactory mitral and tufted cells modulates energy expenditure in a sex-specific manner. a** Schematic of the experiment. Mice were either treated with CNO prior to behavior testing or received saline prior to CNO in indirect calorimetry assays. **b** Immunostaining of MOB from Tbx21-Cre+-hM3Dq+ animal (Cre+), citrine (green) and cFos (red), scale 300 μm, Two independent experiments, $N = 6$. **c** Analysis of cFos positive cells in MOB, males (light blue/blue) and females (pink/red), males, $N = 7$ per group, females, $N = 8$/Cre− and $N = 6$/Cre+, Two-way Anova with Tukey's post hoc comparison. **d** Buried pellet test 20 min following CNO injection, males, $N = 11$/Cre- and $N = 6$/Cre +, females, $N = 9$ per group, Two-way Anova with Tukey's post hoc comparison. **e** Open field test 20 min following CNO injection, $N = 11$ per group, Two-way Anova with Tukey's post hoc comparison. **f** Levels of corticosterone measured 90 min after CNO injection, males, $N = 8$ per group, females, $N = 7$ per group. **g, h** $VO_2$ for 3 h post-CNO Injection, males, $N = 14$ per group, females,

$N = 13$ per group, Two-way Anova with Sidak's post hoc comparison. **i, j** Food intake for 3 h post-CNO Injection, males, $N = 14$ per group, females, $N = 13$ per group. **k, l** Total activity for 3 h post-CNO Injection, males, $N = 14$/per group, females, $N = 13$ p. **m, n** Refeeding after overnight fasting post-CNO Injection, males, $N = 15$/Cre− and $N = 17$/Cre +, females, $N = 12$/Cre− and $N = 10$/Cre +. **o–q** Representative infrared images of ROI selected to measure BAT temperature and analysis of BAT temperature upon CNO injection, males, $N = 8$/Cre− and $N = 11$/Cre +, females, $N = 9$/Cre− and $N = 12$/Cre +, Three-way Anova. **r** Plasma free fatty acid levels in mice collected 90 min post CNO delivery, males, $N = 8$ per group, females, $N = 7$ per group, Two-way Anova with Tukey's post hoc comparison. All bar graphs are presented as mean values ± SEM. Table S2 contains the detailed results of the statistical analysis. *$p < 0.05$, **$p < 0.01$, ***$p < 0.001$. Source data are provided as a Source Data file. corticosterone (CORT), Clozapine-N-Oxide (CNO), Brown adipose tissue (BAT).

this effect was not significantly different in males (Fig. 2e). CNO led to a robust sex-specific induction of CORT levels in female Cre$^+$ mice 90 min after injection (Fig. 2f). Using indirect calorimetry, we observed a female-specific VO$_2$ increase upon OB$^{Tbx21}$ stimulation (Fig. 2g, h). CNO did not alter VO$_2$ in the absence of Cre recombinase and Cre$^+$ mice treated with saline did not demonstrate significant changes in VO$_2$ (Supp. Figure 2b). Analysis of covariance (ANCOVA) of energy expenditure (EE) with body mass as a covariate demonstrated increased energy metabolism in female Cre$^+$ mice upon hM3D stimulation (Supplementary Fig. 2c). Females' VO$_2$ elevation following CNO delivery declined after 3 h and returned to baseline levels (Supplementary Fig. 2d). A stereotaxic strategy to target a restricted number of mitral/tufted cells using a DREADD adeno-associated virus (AAV) resulted in a modest non-significant increase of female VO$_2$ (Supplementary Fig. 2e–g), suggesting that a larger number or different sub-populations of mitral cells are required to drive enhanced energy expenditure. A similar increase in female VO$_2$ was observed when CNO was administered for 3 consecutive days (Supplementary Fig. 2h). No changes in food intake and total activity were observed upon CNO injection in both sexes (Fig. 2i–l). To further investigate the effect of OB$^{Tbx21}$ stimulation on feeding, we measured refeeding after an overnight fast and observed a minor but significant elevation in caloric intake in female Cre$^+$ mice over Cre$^-$ controls, which was not observed in males (Fig. 2m, n). CNO injection was associated with a female-specific increase in BAT temperature in Cre$^+$ mice, peaking 120 min post injection and mirroring the maximum VO$_2$ effect (Fig. 2o–q). Assessment of brown adipocyte thermogenic function showed increased basal mitochondrial oxygen consumption rate (OCR) in Cre$^+$ females compared to Cre$^+$ males (Supplementary Fig. 2i). No difference in gene expression of thermogenic genes was notable in BAT of Cre$^+$ mice over controls, suggesting that the effects on thermogenesis depend on transient stimulation of thermogenesis (Supplementary Fig. 2j), most likely as a consequence of central nervous system (CNS)-driven sympathetic stimulation. Consistent with increased EE and BAT fuel utilization through lipid oxidation, free fatty acid (FFA) levels were elevated following CNO injection in Cre$^+$ females over other genotypes of both sexes (Fig. 2r).

## Gonadectomy does not block the metabolic output of activating female mitral and tufted olfactory cells

Sexual dimorphism in regards to olfactory abilities is well established in mice, including sex differences in olfactory detection repertoires[31]. Removal of sex hormones through gonadectomy was shown to blunt the sex specific effects on odor-evoked glomerular exocytosis responses, suggesting that estrogens might play an important role in the tuning of olfactory responses[32]. Given these observations and the link between estrogens and energy metabolism[33,34], we considered that Cre$^+$ females' specific gain in energy expenditure might result from estrogens' effects on olfactory performance. To interrogate the role of gonadal hormones in mediating the sex-specific effects associated with OB$^{Tbx21}$ stimulation, we employed female ovariectomy (OVX) or sham surgery control animals (SHAM) and monitored the animals using indirect calorimetry (Fig. 3a). Surgery success was confirmed through uterus organ weighing at the end of experiment (Fig. 3b). As anticipated, OVX mice gained more weight than SHAM animals, characterized by higher fat mass without affecting lean mass 5 weeks following surgery (Fig. 3c–e). However, ovariectomy did not impair the higher female VO$_2$ and associated EE produced by OB$^{Tbx21}$ neuronal activation (Fig. 3f, g). Both food intake and activity remained similar in female Cre$^+$ mice over Cre$^-$ controls of both SHAM and OVX groups (Fig. 3h, i).

## Sexually dimorphic cFos expression in the MeA and DMH in response to chemogenetic stimulation of olfactory neurons

To interrogate the neuronal identity of cells encoding the high energy expenditure profile upon olfactory stimulation, we examined cFos expression within the CNS upon OB$^{Tbx21}$ chemogenetic activation. Tissue clearing using the iDISCO protocol followed by light-sheet imaging of the whole mouse brain revealed high cFos expression within the MOB, olfactory cortex regions, amygdala, and hypothalamus (Supplementary Movie 1). cFos immunohistochemistry uncovered female-specific neuronal activation within the MeA and DMH, whereas in the ARC, cFos was induced regardless of sex in Cre$^+$ mice (Fig. 4a–d). Interestingly, a male-specific cFos signal was observed in the LH (Fig. 4c). cFos was also induced in both sexes within the VMH (Supplementary Fig. 3a, b). RNA fluorescent in situ hybridization (RNA-FISH) using RNAScope combined with cFos immunohistochemistry revealed that cFos-expressing DMH neurons were predominantly GABAergic as expressing Vgat, independently of sex (Fig. 4e, h and Supplementary Fig. 4a, b). cFos$^+$-ARC neurons were also enriched in Vgat, whereas LH neurons had a similar proportion of inhibitory and excitatory neurons responding to OB$^{Tbx21}$ stimulation (Supplementary Fig. 4a, b).

Because a subset of GABAergic neurons which express LepR in the DMH are rapidly activated by food cue presentation[27,35], we examined their role in the response to both chemogenetic OB$^{Tbx21}$ and TMT stimuli. RNA-FISH identified a minor proportion of DMH$^{LepR+}$ neurons responding to the olfactory cues (Fig. 4f, h), indicating that this neuronal population is not the primary recipient of olfactory stimuli in the DMH. We found that OB$^{Tbx21}$ stimulation resulted in 14–16% of cFos$^+$ neurons also positive for LepR in the DMH and a similar proportion in the LH (Supplementary Fig. 5a, b). DMH$^{LepR+}$ neurons expressing cFos in OB$^{Tbx21}$-stimulated mice were localized to the compact and ventral DMH areas (Fig. 4f). Interestingly, a higher proportion of cFos neurons in the ARC (50–60%) were also LepR positive (Supplementary Fig. 5a, b). ARC$^{LepR}$ neurons strongly modulate the homeostatic control of food intake and energy expenditure, due to their expression in POMC and AgRP neurons[36,37]. Similar cFos activation of LepR$^+$ neurons in ARC and DMH were observed in TMT and OB$^{Tbx21}$ chemogenetic assays with no sex difference observed, whereas in the LH, TMT elicited an increased cFos$^{LepR+}$ response in females compared to males (Supplementary Fig. 5b).

The MeA is heavily involved in the regulation of sexually dimorphic social behaviors, harboring sexual divergence at the synaptic and molecular level[38,39], including sex differences in the identity of GABAergic neurons[40]. Remarkably, we found that a higher proportion of Vgat$^+$ neurons responded to OB$^{Tbx21}$ stimulation in females compared to males in the MeA (Fig. 4g, i). cFos neurons within the MeA were comprised of a combination of excitatory and inhibitory neurons. Taken together, our results indicate that the sexually dimorphic regulation of energy expenditure observed upon OB$^{Tbx21}$ stimulation may be the product of selectively recruiting MeA$^{Vgat+}$ neurons and an unknown population of DMH neurons in the female brain.

## CRH neurons respond to OB$^{Tbx21}$ activation in a sex-specific manner

Because of the observed glucocorticoid increase upon predator smell and OB$^{Tbx21}$ stimulation in our results, we investigated the specific recruitment of CRH neurons in the CNS. CRH neurons in the PVH are recognized as the major category of neurons encoding hormonal responses to stress, by eliciting neuroendocrine responses through the HPA axis[41]. Briefly, parvocellular PVH$^{CRH}$ neurons release CRH in the median eminence where they activate the pituitary gland, thereby stimulating the synthesis and release of adrenocorticotropic hormone (ACTH). Circulating ACTH acts on receptors in the adrenal cortex, triggering glucocorticoid synthesis and secretion[5]. In addition, other CRH-expressing neuronal populations such as central amygdala (CeA) CRH neurons have been associated with non-HPA axis-CRH production, and implicated in the generation of anxiety-like behavior[42,43]. We therefore focused on the

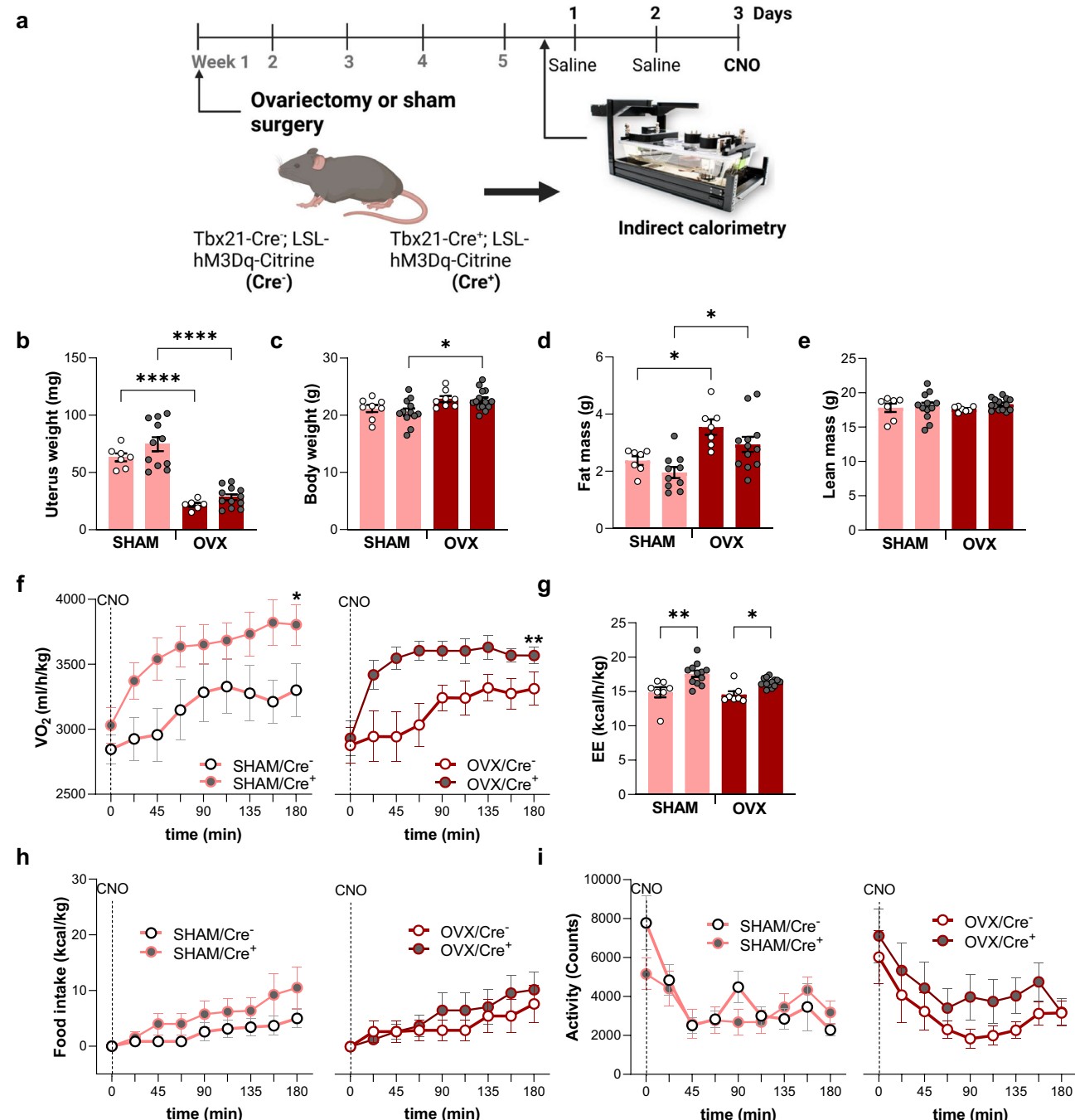

**Fig. 3 | Effect of sex hormones on female energy expenditure driven by chemogenetic activation of olfactory mitral and tufted cells. a** Experimental design, Tbx21-Cre[+]-hM3Dq[+] and control mice received ovariectomy (OVX) or sham surgery (SHAM) 5 weeks prior to indirect calorimetry. **b** Uterus weight at the end of the experiment, pink (SHAM), brown (OVX), SHAM, $N = 7$/Cre[−] and $N = 11$/Cre[+], OVX, $N = 6$/Cre[−] and $N = 10$/Cre[+], Two-way Anova with Tukey's post hoc comparison. **c** Body weight at week 5 post ovariectomy, SHAM, $N = 8$/Cre[−] and $N = 12$/Cre[+], OVX, $N = 8$/Cre[−] and $N = 14$/Cre[+], Two-way Anova with Tukey's post hoc comparison. **d,e** Fat and lean mass at week 5 after ovariectomy, $N = 7–12$/group, Two-way Anova with Tukey's post hoc comparison. **f** VO$_2$ post-CNO injection in SHAM (left) and OVX (right), SHAM, $N = 7$/Cre[−] and $N = 12$/Cre[+], OVX, $N = 8$/Cre[−] and $N = 13$/Cre[+], Mixed-effects model. **g** Energy expenditure (EE) post-CNO, SHAM, $N = 7$/Cre[−] and $N = 12$/Cre[+], OVX, $N = 8$/Cre[−] and $N = 13$/Cre[+], Two-way Anova with Tukey's post hoc comparison. **h** Food intake post-CNO injection in SHAM (left) and OVX (right), SHAM, $N = 7$/Cre[−] and $N = 12$/Cre[+], OVX, $N = 8$/Cre[−] and $N = 13$/Cre[+]. **i** Activity post-CNO injection in SHAM (left) and OVX (right), SHAM, $N = 7$/Cre[−] and $N = 12$/Cre[+], OVX, $N = 8$/Cre[−] and $N = 13$/Cre[+]. All bar graphs are presented as mean values ± SEM. Table S3 contains the detailed results of the statistical analysis. *$p < 0.05$, **$p < 0.01$, ****$p < 0.0001$. Source data are provided as a Source Data file.

ability of both PVH[CRH] and CeA[CRH] neurons to respond to aversive olfactory stimuli in both sexes. Using TMT or control odor, we analyzed cFos localization in the PVH and CeA. cFos was strongly induced in the PVH upon TMT odor in both sexes compared to control odor (Fig. 5a, b), yet Crh mRNA-expressing neurons were weakly recruited in this response (Fig. 5c). Similarly, in the CeA, a large number of neurons responded to TMT, but a small fraction of CRH neurons was activated by TMT (Fig. 5d–f). It should be noted that in females, there was a trend toward an increase in the number of CeA[CRH] neurons responding to TMT (Fig. 5f). We then investigated the ability of OB[Tbx21] stimulation to trigger PVH[CRH] and CeA[CRH] activity (Fig. 5g, h). This stimulation led to massive neuronal activation in the

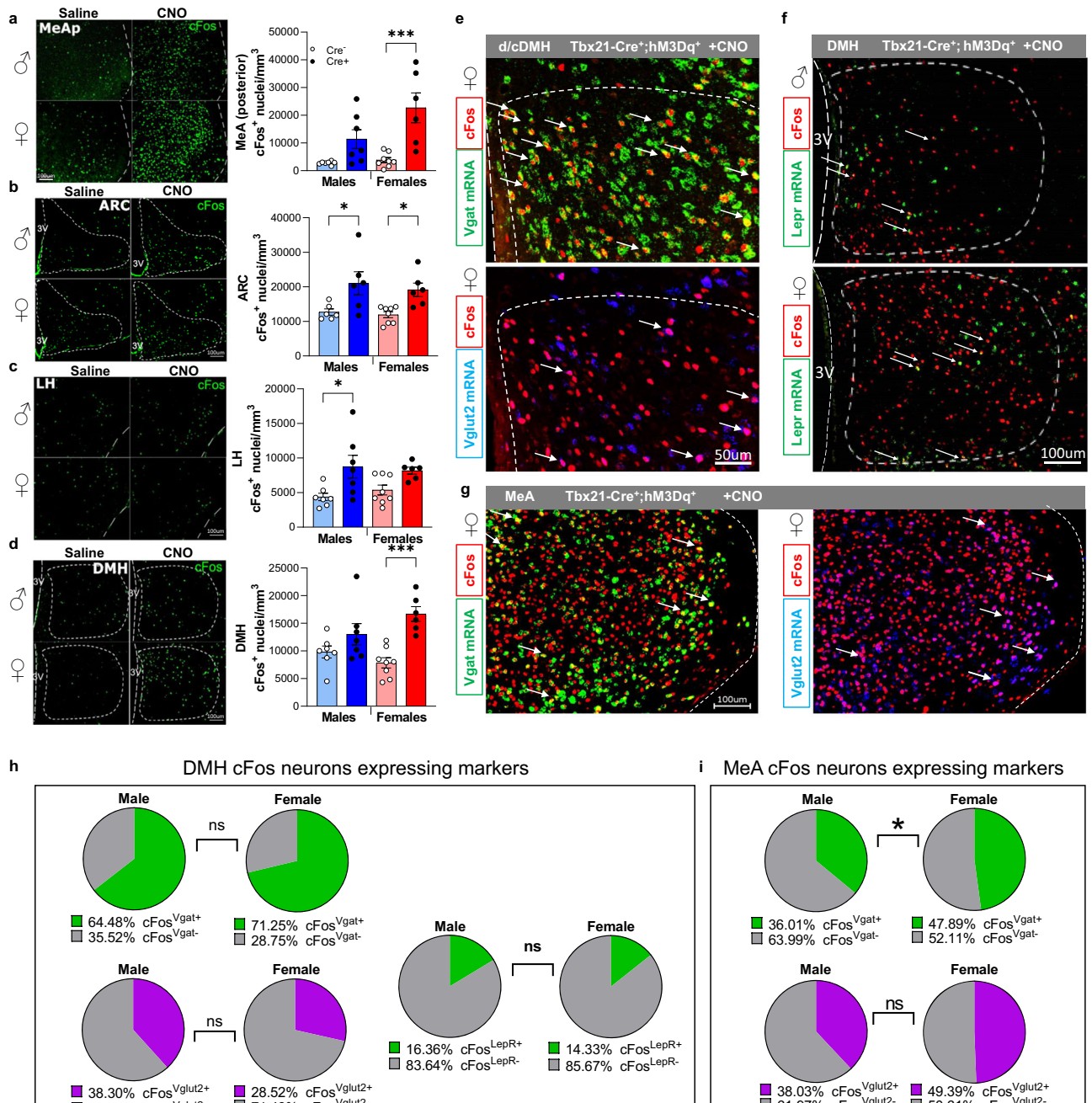

**Fig. 4 | Mediobasal hypothalamus and medial amygdala neuronal activity in response to chemogenetic stimulation of olfactory mitral and tufted cells.** **a**–**d** cFos Immunostaining (green, scale 100 μm) and quantification in posterior medial amygdala (MeAp), ARC, LH, DMH of Tbx21-Cre⁺-hM3Dq⁺ male and female mice, for MeAp, LH, DMH, males, $N = 7$ per group, females, $N = 8$/Cre− and $N = 6$/ Cre + , for ARC, males, $N = 6$ per group, females, $N = 8$/Cre− and $N = 6$/Cre + , Two-way Anova with Tukey's post hoc comparison. **e** Representative image of immunohistochemical and in situ hybridization co-staining in female dDMH for Vgat (green), Vglut2 (blue) and cFos protein (red), yellow cells express Vgat/cFos (top), magenta cells express cFos/Vglut (bottom), scale 50 μm. **f** Representative image of immunohistochemical and in situ hybridization co-staining in DMH for Lepr (green) and cFos protein (red), arrows point to colocalization of signal in neurons, scale

100 μm. **g** MeA immunohistochemical and in situ hybridization co-staining of Vgat (green), Vglut2 (blue) and cFos protein (red), yellow cells express Vgat/cFos, magenta cells express cFos/Vglut, scale 100 μm. **h** Quantification of DMH colocalization between cFos/Vgat and cFos/Vglut2 in male and female mice, $N = 5$ per group (left), and quantification of DMH cFos/Lepr colocalization, $N = 5$ per group (right). **i** Quantification of MeA cFos expression in inhibitory neurons (upper panel) and excitatory neurons (lower panel), $N = 5$ per group, Unpaired $t$-test,Two-tailed. All bar graphs are presented as mean values ± SEM. Table S4 contains the detailed results of the statistical analysis. *$p < 0.05$, ***$p < 0.001$. Source data are provided as a Source Data file. arcuate nucleus of the hypothalamus (ARC), dorsomedial hypothalamus (DMH), lateral hypothalamus (LH) and posterior medial amygdala (MeAp).

female PVH, with CRH neurons showing a robust response to stimuli (Fig. 5i, j). A non-significant increase was observed in males. We observed higher cFos induction in the CeA of OB^Tbx21 activated mice of both sexes, yet CeA^CRH neurons failed to respond to the

chemogenetic stimulus (Fig. 5k, l). These combined observations indicate that PVH^CRH neurons are activated by anxiety-causing olfactory stimuli in the female brain, which could trigger a surge in circulating CORT.

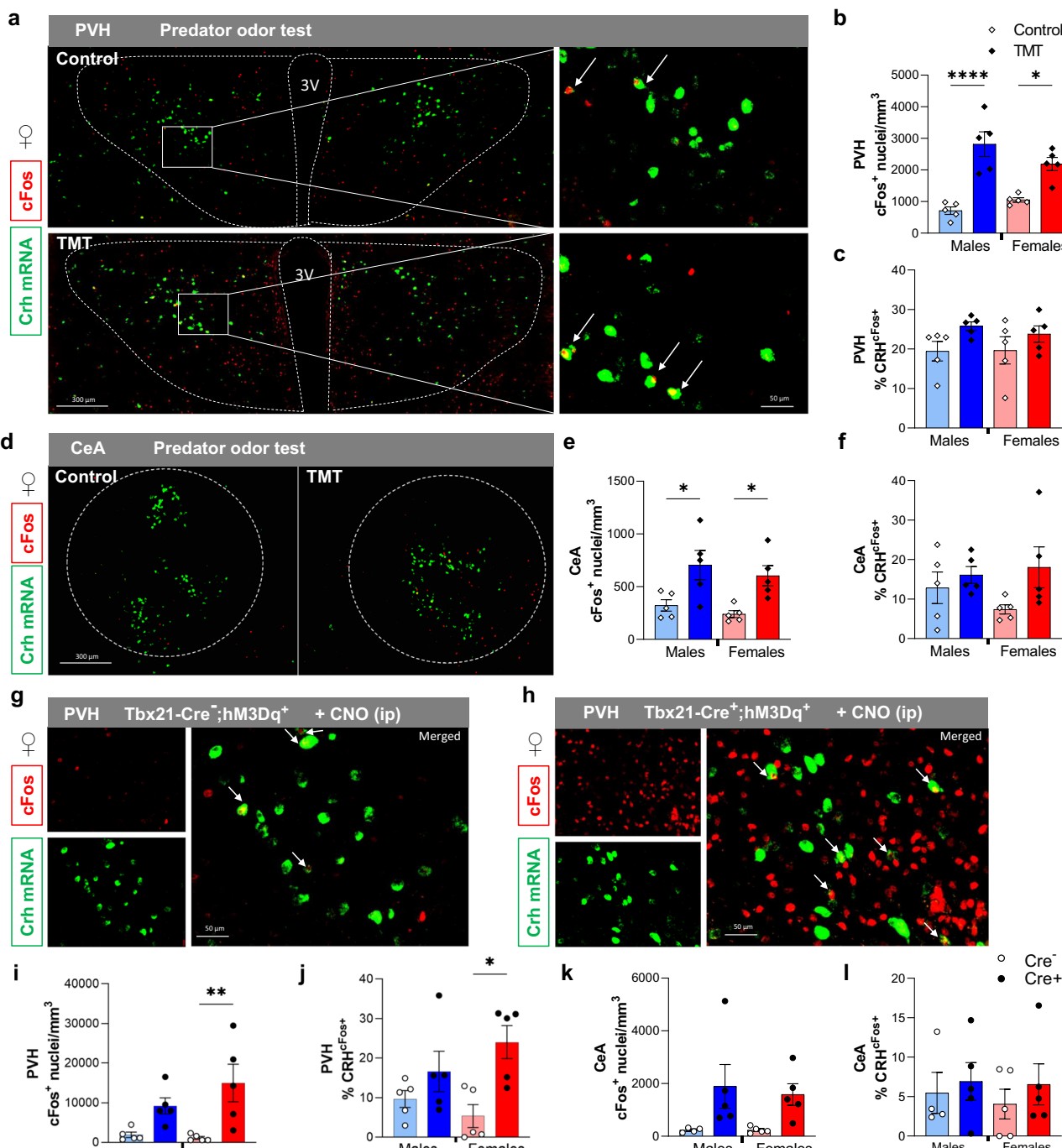

**Fig. 5 | Effect of predator odor exposure and chemogenetic activation of olfaction on activity of CRH-expressing neurons in PVH and CeA.**
**a** Immunohistochemical and in situ hybridization co-staining of Crh (green) and cFos protein (red) in PVH of female mice after exposure to TMT, yellow cells express Crh/cFos, scale 50 μm and 100 μm. **b** Quantification of neuronal activity in PVH after exposure to TMT, males (light blue/blue) and females (pink/red), N = 5 per group, Two-way Anova with Tukey's post hoc comparison. **c** Quantification of the percentage of activated Crh-positive cells in PVH after TMT exposure, N = 5 per group. **d** Immunohistochemical and in situ hybridization co-staining of Crh (green) and cFos protein (red) in CeA of female mice after exposure to TMT, yellow cells express Crh/cFos, scale 300 μm. **e** Quantification of neuronal activity in CeA after exposure to TMT, N = 5 per group, Two-way Anova with Tukey's post hoc

comparison. **f** Quantification of the percentage of activated Crh-positive cells in CeA after TMT exposure, N = 5 per group. **g, h** Immunohistochemical and in situ hybridization co-staining of Crh (green) and cFos protein (red) in PVH of Tbx21-Cre⁻ and Cre⁺-hM3Dq⁺ female mice after CNO injection, yellow cells express Crh/cFos, scale 50 μm. **i, j** Quantification of cFos-positive cells in PVH and CeA post CNO injection, N = 5 per group, Two-way Anova with Tukey's post hoc comparison. **k, l** Quantification of the percentage of activated Crh-positive cells in PVH and CeA, males, N = 4/Cre- and N = 5/Cre + , females, N = 5 per group. All bar graphs are presented as mean values ± SEM. Table S5 contains the detailed results of the statistical analysis. *p < 0.05, **p < 0.01, ****p < 0.0001. Source data are provided as a Source Data file. 2,4,5-trimethylthiazoline (TMT), paraventricular hypothalamus (PVH), central amygdala (CeA).

### The DMH receives afferent projections from mitral and tufted olfactory neurons of the MOB

To characterize the neurons responsible for stress-induced hyperthermia, we mapped out connections of OB^Tbx21 neurons by anterograde tracing using two different viral strategies. First, we assessed anterograde projections of OB^Tbx21 neurons in the synaptically connected regions using the tracer AAV-hSyn-DIO-mCherry

delivered to the MOB of Tbx21-Cre mice. We observed mCherry signal in mitral/tufted cells, and downstream in olfactory cortex (OC): the lateral olfactory tract (LOT), piriform cortex (PIR) and olfactory tubercula (OT), but there were no direct projections to the MBH (Fig. 6a). We then used the polysynaptic tracer AAV1-hSyn-Cre delivered to the MOB of Ai14 (LSL-TdTomato reporter) mice as this virus transduces presynaptic neurons effectively and specifically

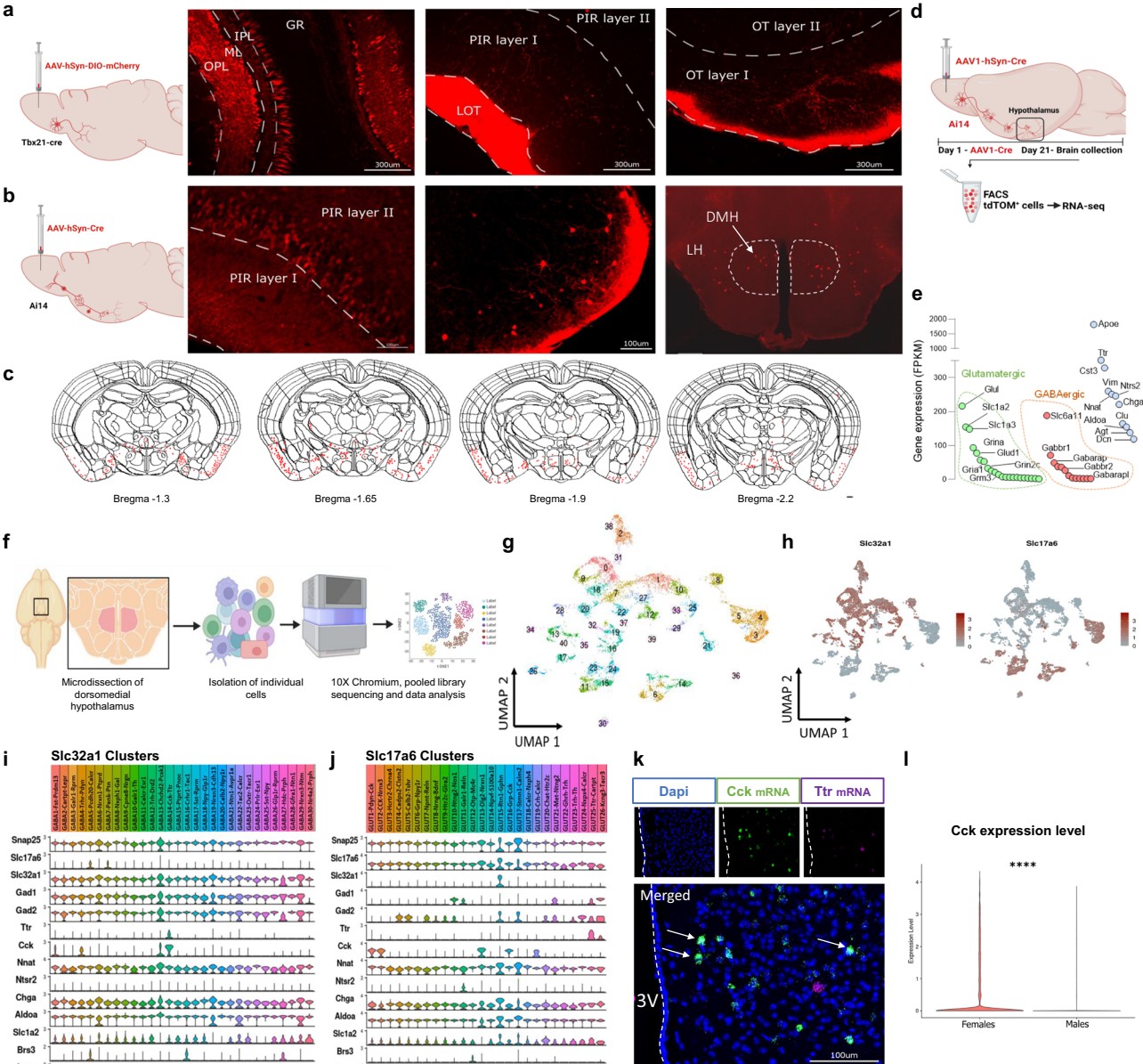

**Fig. 6 | Indirect synaptic connections allow neuronal communication between MOB and DMH nuclei. a** Approach to identify direct synaptic connections with MOB using the presynaptic tracer AAV-hSyn-DIO-mCherry delivered to the MOB of Tbx21-Cre⁺ mice. mCherry positive cells were solely present in olfactory processing centers, Two independent experiments, N = 6. **b** Approach to identify indirect synaptic connections with MOB using the polysynaptic tracer AAV-hSyn-Cre injected into MOB of Ai14 mice (tdTomato reporter) and identification of projection neurons in DMH and LH, Two independent experiments, N = 5. **c** Summary of tdTomato signal localization in brain sections using QuckNII alignment (Allen Brain Atlas reference), Two independent experiments, N = 5. **d** Cartoon representation of cell sorting analysis to collect hypothalamic TdTomato-positive neurons and RNA-sequencing. **e** tdTomato+ hypothalamic neurons transcriptome levels of GABAergic, glutamatergic and other enriched genes, Two independent experiments, N = 14. **f** Experimental design to isolate single cells from microdissected DMH and

scRNA-seq. **g** Uniform Manifold Approximation and Projection (UMAP) clustering of neuronal populations in the DMH identified by single cell RNA sequencing (10, 028 neurons, 6 pooled mice, 41 clusters). **h** UMAP highlighting GABAergic (*Slc32a1*) and glutamatergic (*Slc17a6*) neuronal clusters. **i,j** Subclustering into inhibitory and excitatory clusters revealed cluster GABA 14 as one coexpressing Ttr and Cck, color scheme of clusters corresponds to color scheme of UMAPs in Supplementary Fig. 7 **k** Immunohistochemical and in situ hybridization co-staining of Cck (green) and Ttr mRNA (magenta) in DMH, white represent Cck/Ttr co-expression, scale 100 μm. **l** Violin plot of Cck expression in DMH neurons, Model-based Analysis of Single-cell Transcriptomics (MAST) analysis. Table S6 contains the detailed results of the statistical analysis. ****p < 0.0001. Source data are provided as a Source Data file. main olfactory bulb (MOB), dorsomedial hypothalamus (DMH), lateral hypothalamus (LH).

drives Cre-dependent transgene expression selectively in post-synaptic neuronal targets, allowing polysynaptic anterograde tracing[44]. Strong expression of tdTomato was detected in MOB with a large number of positive tdTomato cells observed in PIR, OT and MeA (Fig. 6b, c). Expression of tdTomato was also observed in the anterior hypothalamic nucleus (AHN, Bregma −1.65), DMH (Bregma −1.9), LH (Bregma −1.9 and −2.2), and posterior hypothalamic nucleus (PHN, Bregma −2.2). We did not observe tdTomato-expressing neurons in the PVH. We confirmed that tdTomato fluorescence was specific to neuronal cells, as colocalizing with NeuN and absent in GFAP+ astrocytes (Supplementary Fig. 6a, b). Because of our cFos neuronal activity mapping assays (Fig. 4), we then focused on DMH neurons as putative regulators of energy expenditure. Fluorescence-activated cell sorting (FACS) followed by sequencing of TdTomato-expressing cells from the microdissected hypothalamus revealed enrichment of transcripts including Ttr, Nnat, Ntsr2 and Clu (Fig. 6d, e). Both GABAergic and glutamatergic markers were also present among the enriched genes, in line with our observations regarding cFos expression upon OB[Tbx21] stimulation.

We then performed single-cell RNA-sequencing (scRNA-seq) of the microdissected DMH to evaluate the identity of cells expressing the enriched genes from the FACS analysis (Fig. 6f). We profiled the gene expression of 10,028 neurons from male and female C57BL/6 animals using scRNA-seq. Unsupervised clustering of these data revealed 41 neuron types (Fig. 6g). We further clustered neurons inhibitory (expressing Slc32a1) and excitatory (expressing Slc17a6) neurons separately to reveal cellular complexity in the DMH (Fig. 6h, Supplementary Fig. 7a–d, Supplementary Fig. 8a, b). Furthermore, we

analyzed the presence of the most enriched transcripts from our previous FACS tdTomato analysis in inhibitory and excitatory neuronal clusters (Fig. 6i, j). Many of these transcripts, such as Nnat, Chga, Aldoa, Slc1a2 were enriched in most neuronal clusters, therefore not defining a specific cellular population. However, Ttr expression was restricted to a selective inhibitory cluster (GABA14) and two excitatory clusters (GLUT25 and 26). This particular inhibitory cluster was also abundant in Cck, but did not contain either Brs3 or Lepr, two DMH neuronal chemotypes previously shown to be involved in energy balance[27,45,46]. Using FISH, we confirmed that Cck and Ttr mRNAs were co-expressed in select DMH neurons (Fig. 6k). Remarkably, we also found that Cck expression was higher in female DMH neurons compared to males (Fig. 6l), indicating a potential source of sexual dimorphism occuring at the level of the DMH.

## CCK neurons mediate the increased energy expenditure caused by olfactory neurons' activation
Our results indicate that DMH neurons expressing Cck are likely to transduce olfactory inputs into physiological responses. We employed FISH to determine the presence of Cck in cFos neurons responding to chemogenetic OB[Tbx21] stimulation (Fig. 7a). Cck-expressing neurons were found in the dorsal (dDMH) and compact (cDMH) regions, detectable at Bregma −1.7 mm ("anterior DMH") and Bregma −1.8 mm ("middle DMH"). Chemogenetic OB[Tbx21] stimulation led to moderate activation of male cFos[Cck+] neurons in both cDMH and dDMH neurons. In females, there was a robust recruitment of cFos[Cck+] neurons in the cDMH (Fig. 7a, b). TMT odor exposure was associated with a higher response of cFos[Cck+] neurons in the cDMH over the dDMH in both sexes (Fig. 7c, d). Examining putative sexual dimorphism in the response of

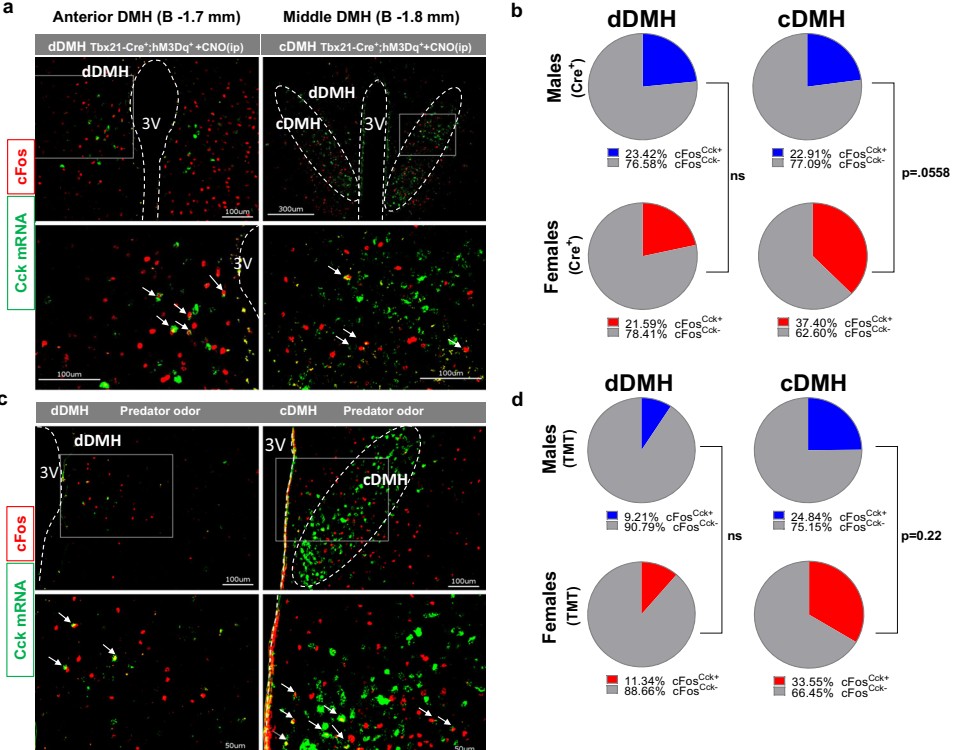

**Fig. 7 | Activation of CCK-expressing neurons in DMH upon chemogenetic stimulation of olfaction or predator odor exposure. a** Representative image of immunohistochemical and in situ hybridization of co-staining for Cck (green) and cFos protein (red) in dDMH and cDMH post-CNO injection, scale 100 μm and 300 μm. **b** Quantification of the percentage of cFos-positive cells that co-express CCK in dDMH (left) and cDMH (right) post-CNO injection, N = 6 per group, Unpaired t-test, Two-tailed. **c** Representative image of immunohistochemical and in situ hybridization of co-staining for Cck (green) and cFos protein (red) in dDMH and cDMH after TMT exposure, scale 50 μm and 100 μm. **d** Quantification of the percentage of cFos-positive cells that co-express CCK in dDMH (left) and cDMH (right) after TMT exposure, N = 4 per group, Unpaired t-test, Two-tailed. Source data are provided as a Source Data file. dorsal dorsomedial hypothalamus (dDMH) and compact dorsomedial hypothalamus (cDMH), clozapine-N-Oxide (CNO), 2,4,5-trimethylthiazoline (TMT).

 

cDMH^[Cck+] neurons to TMT predator odor revealed a trend toward higher activation in female animals compared to males, yet not significantly increased (Fig. 7d).

Previously, a population of DMH neurons containing Cck and negative for LepR, was shown to be selectively activated by refeeding following prolonged fasting with their stimulation inhibiting feeding[47]. Using scRNA-seq, we confirmed the existence of the DMH^[Cck] and DMH^[Lepr] neuronal chemotypes, which mark different inhibitory neuronal clusters (Fig. 6i). As our findings point to a female-selective recruitment of cDMH ^[Cck+] neurons in response to stressful olfactory cues, we evaluated the direct role of these neurons in energy expenditure and/or plasma CORT release. We delivered AAV8-DIO-hM3D-mCherry or AAV8-hSyn-DIO-mCherry bilaterally in the d/cDMH of Cck-IRES-Cre mice and confirmed accurate targeting of the region (Fig. 8a, b). cFos induction was visible upon CNO treatment in mCherry-positive neurons found bilaterally in d/cDMH (Fig. 8b). Remarkably, CNO treatment of Cck-hM3D^[+] mice led to a robust increase in VO$_2$ in both males and females (Fig. 8c, d), characterized by a more elevated response in females. CNO injection was not associated with burst of locomotor activity in either males or females, despite Cck-hM3D^[+] females exhibiting a high level of activity preceding CNO delivery (Fig. 8e). DMH^[CCK] stimulation significantly reduced feeding after an overnight fast in both sexes (Fig. 8f). We observed that CORT levels were not increased upon DMH^[CCK] activation in either sex (Fig. 8g), as opposed to olfactory stimulation assays, indicating that DMH^[CCK] neurons do not mediate the rise in blood glucocorticoids associated with stressful odors. Interestingly, the stimulation of DMH^[CCK] neurons resulted in comparable alterations in energy expenditure to the stimulation of MOB mitral cells in females.

To further explore the recruitment of DMH^[CCK] neurons in mediating the stress-induced thermogenic response associated with stressful odor stimuli, we chronically inhibited synaptic release of DMH^[CCK] neurons using Cre-dependent tetanus toxin virus (AAV-DIO-eGFP-2A-TeNT)[48]. Cck-IRES-Cre mice received AAV-DJ-CMV-DIO-eGFP-2A-TeNT (TeNT) or AAV DJ-CMV-DIO-eGFP (eGFP) in the d/cDMH (Fig. 8h). GFP expression was visible in DMH neurons (Fig. 8i). Three weeks following surgeries, body weights, food intake and activity levels were similar among TeNT and eGFP animals (not shown). We then investigated the impact of silencing DMH^[CCK] neurons on TMT-associated changes in energy expenditure in female mice. Upon TMT odor presentation, eGFP animals had increased VO$_2$ compared to control swab "no odor" (Fig. 8j, k). In TeNT animals, we failed to observe a burst in VO$_2$ upon swab introduction in the "no odor" condition which is present in eGFP controls. Remarkably, in TeNT-treated animals, the VO$_2$ gain associated with TMT exposure remained at a level comparable to eGFP animals which were exposed to a "no odor" condition (Fig. 8j, k). Yet, even though the magnitude of the TMT response was smaller than in eGFP control animals, an increased stress-induced gain in energy expenditure could still be observed in TeNT-treated mice. Remarkably, silencing DMH^[CCK] neurons did not impact the amplitude of the increased locomotor activity associated with TMT perception (Fig. 8l, m), indicating that these neurons are unlikely to mediate locomotive changes associated with predator odor perception. Importantly, we observed that silencing DMH^[CCK] neurons restored feeding associated with overnight fast to no odor/eGFP levels despite the presence of TMT (Fig. 8n, o). Taken together these results demonstrate that DMH^[CCK] neurons participate in the generation of thermogenic and feeding responses driven by the detection of certain predator threats.

## Discussion

Whether select olfactory cues can regulate changes in energy metabolism has long remained an unanswered question. Using a combination of odorant presentation and chemogenetic activation of mitral cells of the MOB, we provide evidence that stressful olfactory stimuli promote a female selective increase in energy expenditure. We also show that TMT odor exposure is associated with the suppression of feeding following an overnight fast. We also demonstrate that CCK-expressing neurons in the DMH are activated by these cues, and their neuronal manipulation through chemogenetic stimulation and neuronal silencing further confirms a role for these neurons in promoting stress-induced thermogenesis and feeding suppression upon predator odor detection.

Our observations regarding the role of DMH^[CCK] neurons in feeding are in line with a previous report in which a DMH neuronal chemotype, expressing Cck among other genes, suppressed feeding[47]. We demonstrate that selective activation of DMH^[CCK] neurons inhibits food intake, whereas their neuronal silencing boosted feeding following an overnight fast (Fig. 8n, o). scRNA-seq of the DMH reveals that DMH^[CCK] neurons constitute a segregated neuronal population from previously characterized DMH^[LepR] or DMH^[Brs3] neurons[27,45,46], therefore highlighting the heterogeneity and complexity of the DMH in the regulation of energy balance. We also expose the sex-specific changes in Cck expression found in these neurons, where enriched Cck expression found in females is associated with more pronounced gain in thermogenesis upon chemogenetic stimulation.

There were similarities in the metabolic and cellular outputs to TMT and OB^[Tbx21] stimulation, including a gain in female energy expenditure, however with different kinetics, which might be attributed to the long-lasting effects of CNO on neuronal activation, as opposed to more transient responses generated by a natural odorant (Figs. 1 and 2). A remarkable difference in these two models was their divergence in controlling food intake, as feeding was not inhibited in OB^[Tbx21] stimulation in the refeeding post fast experiment, whereas TMT was associated with an inhibitory action on refeeding in the fasted state. Because our olfactory chemogenetic approach stimulated most mitral cells of the OB, confounding signals arising from multiple olfactory pathways are likely to be generated and alter the central regulation of feeding, with females being more prompt to find a buried pellet test and consuming more food in the fasted-refed state. CORT levels were also more elevated in female OB^[Tbx21] stimulation as compared to TMT, and glucocorticoids are well known to increase feeding[9]. It is well established that CORT induces hyperphagia and increases fat mass when administered chronically[10,11]. Due to the nature of the DREADD stimulus, it is plausible that CORT levels remained elevated for a prolonged time after CNO injection and had modulatory action of food intake on OB^[Tbx21]-stimulated animals. Alternatively, the difference between TMT and chemogenetic OB stimulation models could lie in additional recruitment of the trigeminal system in the case of TMT exposure. Remarkably, TRPA1-mediated nociception has been shown to play a crucial role in predator odor-evoked innate fear/defensive behaviors, including TMT[49]. Nociceptive TRPA1 detection of TMT in the nasal cavity through non-olfactory pathways could contribute to additional modulation of defensive behavior, including suppression of feeding despite prolonged fasting.

The cellular resolution of stress-responding neurocircuits has heavily implicated hypothalamic paraventricular CRH neurons in the endocrine control of the stress response, regulating complex behaviors following stress, including grooming, rearing or walking[2]. It has already been established in fiber photometry experiments that PVH^[CRH] neurons respond to TMT odor exposure, before the onset of freezing episodes[4]. Yet, our cFos immunolabelling assays failed to measure CRH-dependent response to TMT, while we observed that non-identified PVH neurons were activated by this stimulus (Fig. 5). This discrepancy could be caused by rapid changes in calcium dynamics within CRH neurons, which might not be captured using cFos as a proxy for neuronal activation. Yet, we observe that OB^[Tbx21] stimulation leads to a female selective stimulation of PVH^[CRH] neurons, indicating that olfactory "stress" recruits the canonical stress response and might account for CORT release upon CNO injection. Another important aspect of the

 

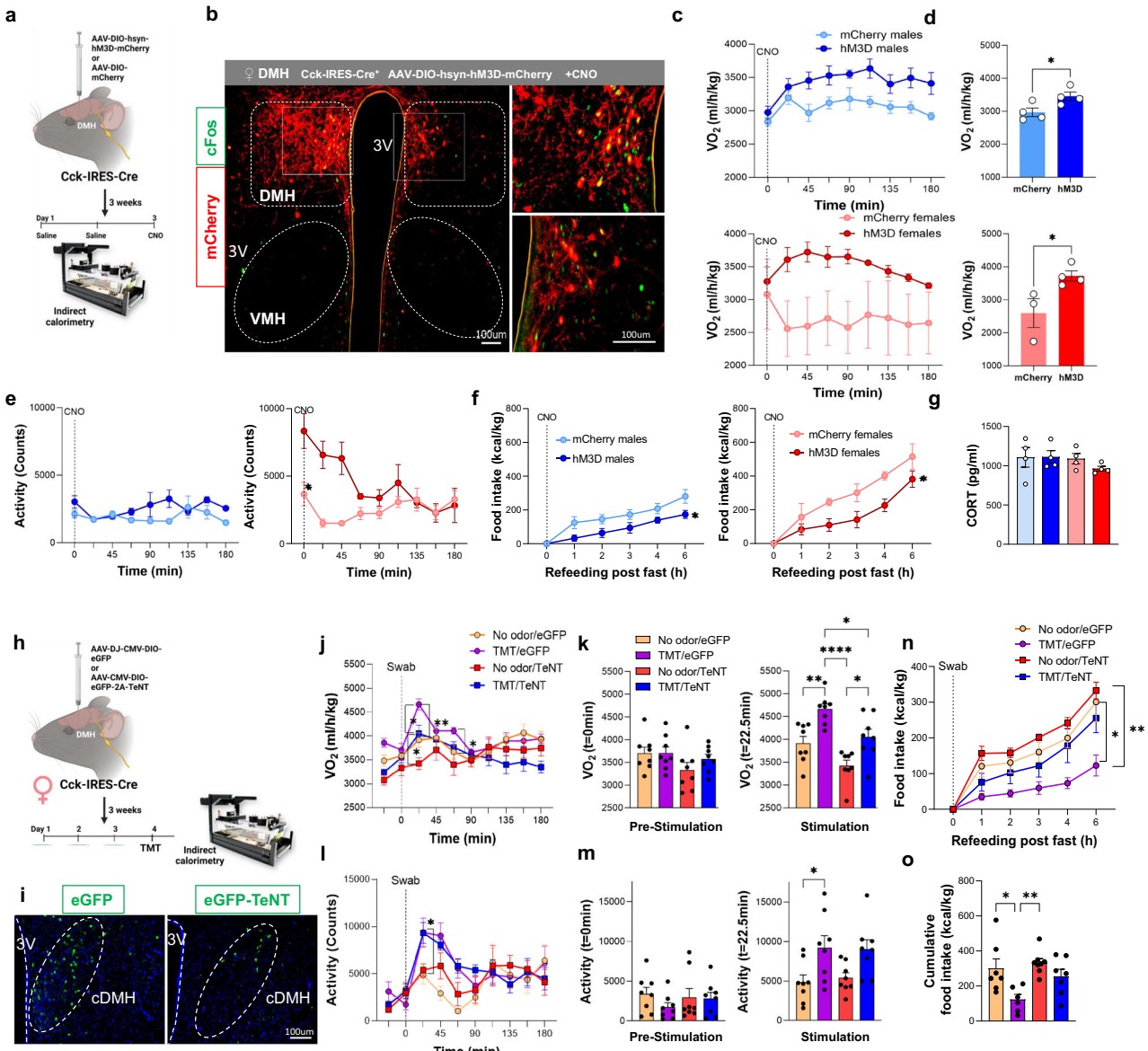

**Fig. 8 | Chemogenetic activation and tetanus toxin-mediated synaptic silencing of CCK-expressing neurons in DMH. a** Experimental design for chemogenetic manipulation of CCK-expressing neurons in the DMH. **b** Bilateral stereotaxic targeting of DMH neurons using AAV-DIO-hM3D-mCherry shows robust mCherry signal in the d/cDMH overlapping with cFos after CNO injection, Two independent experiments, $N = 15$. **c** Oxygen consumption in CCK-IRES-Cre⁺ mice after CNO injection, males, $N = 4$ per group, females, $N = 3$/mCherry and $N = 4$/hM3D. **d** Average VO₂ in CCK-IRES-Cre⁺ measured for 3 h post-CNO injection, males, $N = 4$ per group, females, $N = 3$/mCherry and $N = 4$/hM3D, Unpaired $t$-test,Two-tailed. **e** Total activity in CCK-IRES-Cre+ mice after CNO injection, males, $N = 4$ per group, females, $N = 3$/mCherry and $N = 4$/hM3D. **f** Food intake measured after overnight fasting and upon CNO injection, males, $N = 4$ per group, females, $N = 3$/mCherry and $N = 4$/hM3D, Two-way Anova with Sidak's post hoc comparison. **g** CORT levels post-CNO injection, $N = 4$ per group. **h** Experimental design for tetanus toxin-mediated synaptic silencing of CCK-expressing neurons in the DMH. **i** eGFP positive cells in cDMH after stereotaxic AAV injection, Two independent experiments, $N = 16$. **j** VO₂

measured upon TMT or control swab exposure, $N = 8$ per group, Three-way Anova. **k** VO₂ before the stimulus (t = 0 min) and after exposure to the stimulus (t = 22.5 min), $N = 8$ per group, Two-way Anova with Tukey's post hoc comparison. **l** Activity measured upon TMT or control swab exposure, $N = 8$ per group, Three-way Anova. **m** Activity before the stimulus (t = 0 min) and after exposure to the stimulus (t = 22.5 min), $N = 8$ per group, Two-way Anova with Tukey's post hoc comparison. **n** Food intake measured after overnight fasting and upon TMT or control swab exposure, eGFP, $N = 7$/No odor and $N = 6$/TMT, TeNT, $N = 8$/No odor and $N = 7$/TMT, Three-way Anova. **o** Cumulative food intake after 6 h of refeeding under TMT or control swab exposure, eGFP, $N = 7$/No odor and $N = 6$/TMT, TeNT, $N = 8$/No odor and $N = 7$/TMT, Two-way Anova with Tukey's post hoc comparison. All bar graphs are presented as mean values ± SEM. Table S7 contains the detailed results of the statistical analysis. *$p < 0.05$, **$p < 0.01$, ****$p < 0.0001$. Source data are provided as a Source Data file. Clozapine-N-Oxide (CNO), 2,4,5-trimethylthiazoline (TMT), corticosterone (CORT), compact dorsomedial hypothalamus (cDMH), Adeno-associated viruses (AAV).

exposure to predator odor is the suppression of feeding responses, and whether CRH neurons contribute to this behavior is an important question. Previous studies have established that chemogenetic inhibition or removal of CRH neurons did not modulate food intake or body weight in mice[50,51], yet conflicting evidence has associated PVH^CRH neurons with the regulation of feeding by glucagon-like peptide-1 (GLP-1)

and Neuropeptide Y (NPY) signaling[52,53]. Here our data clearly demonstrate a crucial role for DMH^CCK neurons in the modulation of stress-induced suppression of feeding (Fig. 8). As central delivery of the CRH peptide in rodent models recapitulated the stress hyperthermic response through modulation of unknown DMH neurons[8], these findings combined with our results suggest that PVH^CRH and DMH^CCK

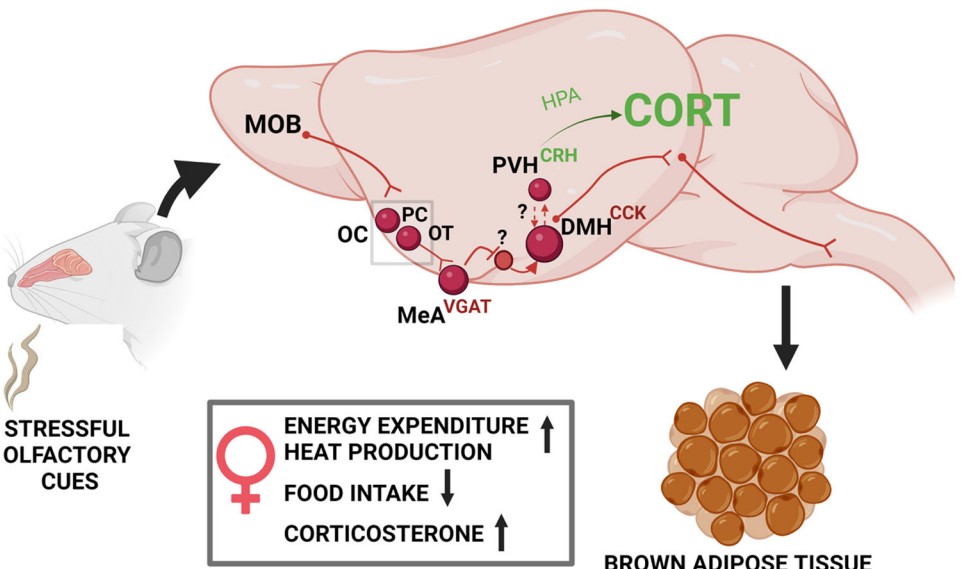

**Fig. 9 | Model for the stress-induced recruitment of thermogenesis by olfactory cues in the female brain.** Stressful cues are detected by olfactory pathways and integrated by the MeA, where increased GABAergic activity in the female brain leads to recruitment of CCK neurons from the DMH and stress-induced thermogenesis through BAT activation and suppression of food intake. Recruitment of CRH neurons in the PVH drives CORT release through HPA axis stimulation, in a process that might depend on the activation of CCK neurons. Cholecystokinin (CCK), corticotropin-releasing hormone (CRH), corticosterone (CORT), dorsomedial hypothalamus (DMH), hypothalamic-pituitary-adrenal (HPA) axis, main olfactory bulb (MOB), medial amygdala (MeA), olfactory cortex (OC), olfactory tubercula (OT), paraventricular nucleus of the hypothalamus (PVH), piriform cortex (PC).

neurons might work jointly to regulate both hyperthermia and feeding suppression upon stress (Fig. 9). Functional neuroanatomy experiments in the rat brain have revealed that the DMH functions as a hub for stress signaling, with monosynaptic projections to the rostral medullary raphe region for sympathetic control of thermogenesis and the PVH for neuroendocrine output[13,54]. Further studies are required to understand the communication between DMH and PVH neuronal populations in the regulation of stress-induced metabolic responses.

Multiple stressors are likely to stimulate parallel neurocircuits to regulate organismal stress responses. For example, physical restraint, but not TMT odor, selectively induced anorexigenic pro-opiomelanocortin (POMC) neuron activity in ARC to suppress feeding[55]. Therefore, our observed results of refeeding inhibition after overnight fast in TMT-exposed mice imply that other neuronal targets, beyond POMC neurons, are required for the suppression of feeding, such as DMH$^{CCK}$ and possibly to a lesser extent DMH$^{LepR}$ cellular populations. Therefore, canonical stress responses encoded by PVH$^{CRH}$ neurons modulate behavior and glucocorticoid release but additional neuronal chemotypes are likely to participate in stress-related changes in energy homeostasis, such as suppressed feeding and increased energy expenditure. A better understanding of the stressful cues integrated by DMH$^{CCK}$ neurons is necessary to dissect divergent neurocircuits integrating stress responses.

The biological relevance of a sexually dimorphic response in the rodent brain remains unclear. In mice, sexually dimorphic differences in olfactory sensory receptor repertoires allow the detection of sex-specific odors[31]. Presentation of a select panel of volatile chemicals with fruity smell evoked more rapid neurotransmitter exocytosis in glomeruli of the MOB in female mice in a concentration-dependent manner[32]. OVX blunted the rapid sex specific effects on odor-evoked glomerular exocytosis responses, indicating that estrogens play an important role in the tuning of female olfactory responses[32]. However, in our results, OVX failed to block the increased energy expenditure profile of OB$^{Tbx21}$ stimulated females (Fig. 3). Although we have not tested the presence of a stress-induced thermogenic response in males lacking gonads and therefore fully examine the role of sex hormones in our models, our findings suggest that this metabolic response is not influenced by the absence of estrogens, but rather might be the consequence of heightened stress integration in the female brain. In rodent models, stressful stimuli tend to have more profound effects in females[56]. Female rats have greater and more persistent CORT responses to stress[57]. Rat parvocellular PVH innervation, measured by the density of synaptophysin staining, is greater in females than in males[23]. In addition, social isolation alters the intrinsic properties of CRH neurons in female, but not in male mice[58]. In humans, there is also some evidence that stress-related disorders might be more prominent in women[22]. We observe sexual dimorphism in the cellular composition of DMH neurons involved in the response, with enhanced female Cck expression, in addition to a selective recruitment of these neurons in the female brain by aversive olfactory stimulation. A plausible sexually dimorphic region controlling the female specific response is the MeA, which also exhibit sex-specific neuronal activation to olfactory stress (Fig. 4). The MeA controls sexually dimorphic social behaviors, and its synaptic organization differs in males and females[38,39]. scRNA-seq of the MeA revealed molecular sex differences in GABAergic neurons, which might account for parenting and infanticide behaviors[40]. We find that a higher proportion of MeA$^{Vgat}$-positive neurons responded to OB$^{Tbx21}$ stimulation in females compared to males (Fig. 4). We postulate that the activation of MeA$^{Vgat}$ neurons might inhibit an unknown neuronal population which normally suppresses the activity of DMH$^{CCK}$ neurons, therefore driving stress-induced thermogenesis (Fig. 9). Better understanding of the nature of GABAergic neurons found in the MeA and their female-specific cue integration are necessary to elucidate the basis of female heightened stress sensitivity.

In conclusion, this study provides insight into sexually dimorphic responses to olfactory stimuli and identifies a female-selective gain of energy expenditure and reduced feeding upon aversive olfactory stimulation. We identified a neuronal population in the DMH expressing Cck, which plays an essential role in integrating these cues and modulating energy balance. This sex-specific neuronal circuit may be a potential therapeutic target for stress and eating disorders.

## Methods

### Animals

All procedures were approved by the Animal Care and Use Committee of Cedars Sinai Medical Center or California Institute of Technology. In this study, we used Ai14 (Jax strain 007914), Tbx21-Cre (Jax strain 024507), R26-LSL-hM3Dq-DREADD (Jax strain 026220), Cck-IRES-Cre (Jax strain 012706), and C57BL/6 J (Jax strain 000664). Mice were bred in our colony according to Jackson Laboratory instructions for each strain. For chemogenetic olfactory activation experiments, we crossed Tbx21-Cre mice (expressing cre-recombinase in the mitral cell layer of the MOB) with R26-LSL-Gq-DREADD mice to allow Cre recombinase-inducible expression of a CAG promoter-driven HA-hM3Dq-pta-mCitrine to conditional stimulate the activity of mitral cells in olfactory bulb upon admission of the inert DREADD agonist CNO. In all experiments, appropriate littermate controls were used. Mice were maintained on a 14 h:10 h light:dark cycle, fed normal chow (PicoLab Rodent 20 5053*, LabDiet). All experiments are done at 22 °C. Both male and female mice were used in this study unless otherwise stated.

### Stereotaxic viral Injection

Anesthesia was induced using isoflurane (induction, 5%; maintenance, 1–2%). Mice were placed on the stereotactic frame (Kopf Instruments). A 0.5 mm hole was drilled into the skull over the target site. All injections were done using a microliter syringe (Hamilton) and UMP3 pump (WPI) mounted directly on the stereotactic frame and the viruses were delivered at a rate of 2 nl sec-1. After the Injection, the needle of the syringe was kept in place for 10 min to reduce backflow and then withdrawn slowly. Carprofen (0.08 ml/kg body weight) and buprenorphine (0.1 mg/kg body weight) were administered subcutaneously before the surgical procedure. Animals were recovered on a heating pad until normal behavior resumed. For the anterograde tracing experiment mice (Tbx21-Cre and Ai14) received AAV-hSyn-DIO-mCherry (Addgene, 50459-AAV8) or AAV.hSyn.Cre.WPRE.hGH (Addgene, 105553-AAV1) bilaterally in each olfactory bulb (300 nl on each target site). The coordinates for the MOB were (relative to bregma) A/P + 4.25 mm, M/L ± 0.5, 1.3 mm and D/V −1.4–1.7 mm. Mice were sacrificed 21 days after viral Injection. For the chemogenetic activation of CCK-expressing neurons and Slc1a3-expressing neurons in DMH, 150 μl of AAV-hSyn-DIO-hM3D(Gq)-mCherry (Addgene, 44361-AAV8) or AAV-hSyn-DIO-mCherry (Addgene, 50459-AAV8) viruses were injected bilaterally in DMH (coordinates relative to bregma: A/P −1.9 mm, M/L ± 0.3 mm and D/V −5.6 mm). For the tetanus toxin-mediated synaptic silencing of CCK-expressing neurons in DMH, 150 μl of AAV-DJ CMV DIO eGFP-2A-TeNT (Standford University GVVC, GVVC-AAV-071) or AAV DJ-CMV DIO eGFP (Standford University GVVC, GVVC-AAV-012) viruses were injected bilaterally in DMH (coordinates relative to bregma: A/P −1.9 mm, M/L ± 0.3 mm and D/V −5.6 mm)Verification of the Injection site is performed by IHC. Only animals with viral expression confirmed on both sites were used for experiments that required bilateral viral injection.

### Odor presentation test

12-week-old naïve wild-type mice (C57BL/6J) were used in this experiment. Prior to the assay, mice were habituated to cotton swabs and peanut butter for 2 days. Clean cotton swab and/or peanut butter were placed in the upper right corner of the mouse cage and removed after 2 h. On the day of the experiment, one side of the cotton swab was immersed in one of the following solutions: mineral oil (Sigma-Aldrich, M5904), 20% peanut butter oil (Trader Joe's Crunchy Salted Peanut Butter in mineral oil, filtered), 10% TMT (BioSRQ, 97-TMT) and opposite-sex urine. Urine is collected on the day of the experiment from 10 different mice from the same background. Mice were transferred to cages with fresh bedding, and the cotton swab was placed in the upper right corner of the cage. After 30 min, the cotton swab was removed and 60 min later, mice were sacrificed and brains were collected.

### Buried pellet test

2 h before the beginning of testing mice were transferred to the procedure room to acclimate. A clean mouse cage was filled up with 8 cm of fresh bedding and 2 g of food pellet (PicoLab Rodent 20 5053, LabDiet) was buried 8 cm underneath the surface of the bedding in the middle of the cage facing the edge and being visible to the experimenter. The position of the pellet was constant throughout the experiment. The main parameter measured in this test is the latency to find the hidden food (pellet). Latency is defined as the time between when the mouse was placed in the cage and when the mouse uncovered the food pellet (contact between mouse nose and the food pellet). The total duration of the test was 10 min and was started 20 min after CNO injection.

### Open field test

20 min after CNO injection mice were moved to the open field arena. Spontaneous locomotor activity was measured using the Photobeam Activity System (San Diego Instruments), which consists of a 40 cm-by-40 cm square chamber with 40 cm high transparent Plexiglass walls and virtually divided into center (30 cm-by-30 cm) and perimeter (5 cm along each wall). Two rings of 16 photobeams and optical sensors surrounded the chamber, and beam breaks captured the mouse's horizontal ambulatory activity and vertical rearing behavior. Each mouse was individually placed in the center of the chamber and left undisturbed to explore the environment for a 60 min period freely. The number of rearing, total activity, and percentage of center activity were recorded. Distance, average speed, resting time and trajectory of each mouse were computed using PAS Reporter and PAS PathView software.

### Refeeding after overnight fast

Overnight fasted naïve mice were assigned a group (TMT or control swab) in separate rooms, where they were allowed to acclimate for 2 h. Both normal chow and a cotton swab with no odor (control) or TMT were introduced in the cage in opposite corners. Food intake was measured for 6 h. For OB$^{Tbx21}$ and DMH$^{CCK}$ stimulation using DREADD, mice received CNO 20 min prior to food introduction within the cage. For DMH$^{CCK}$ silencing using tetanus-toxin virus overnight, fasted mice were exposed to a control swab, and 1 week after, they were fasted again and exposed to TMT. Mice in this experiment had previous exposure to TMT.

### Indirect calorimetry

Indirect calorimetry (VO$_2$, EE, RER), physical activity and food intake were collected in an automated home cage eight-chamber phenotyping system (Phenomaster v.6.6.1., TSE). In CNO-treated mice, animals were acclimaed for the first 2 days and were receiving saline (i.p) and on day 3, mice received Clozapine-N-Oxide (CNO, HelloBio, HB6149) at 1 mg/kg body weight, i.p. All injections were done at the same time (11am). Sampling was every 18 min, with 2 min timeframe in each chamber. For TMT-treated mice, mice were acclimated to an odorless swab (mineral oil) for 3 days and on the 4th day, a cotton swab with 10% TMT (BioSRQ, 97-TMT) was or control swab (mineral oil) was placed in the metabolic cage for measuring metabolic activity.

### Quantitative thermal imaging of BAT

BAT temperature was measured by IR camera (FLIR E54, FLIR System) in 12-week-old Tbx21-Cre-/ R26-LSL-hM3Dq-DREADD+ or Tbx21-Cre +/ R26-LSL-hM3Dq-DREADD+ mice. Mice were allowed to acclimate for 3 h to the room where experiments were performed. Measurements were done for 3 consecutive days. Baseline timepoint (0 min) imaging was performed and CNO (1 mg/kg body weight, i.p.) was injected. For TMT exposure naïve C57BL/6 J were used and measurements were done over 1 day: baseline timepoint (0 min) imaging was performed and TMT was introduced in the cage. Unshaved skin temperature in

the interscapular regions was used as a measure of BAT temperature and for each time point, 3 pictures per mouse were collected. Analysis of images was done by a blinded observer using FLIR Tools Thermal Analysis and Reporting Software v.5.13.18031.2002.

## Quantitative PCR analysis

RNA was isolated using trizol/chloroform extraction and RNEasy Plus Micro Kit (Qiagen, 74034). Gene expression was assessed by qPCR Power SYBR™ Green RNA-to-CT™ 1-Step Kit (Applied Biosystem, 4389986). 25 ng of total RNA was used for each reaction, and 18 s gene was used as control. Primers sequences are available in our previous study[59].

## Plasma free fatty acid quantification

Plasma was collected 90 min after CNO injection. Free fatty acid was quantified from plasma by Free Fatty Acid Quantitation Kit (Sigma-Aldrich, Cat. #MAK044), following manufacturer instructions.

## Plasma corticosterone quantification

Blood was collected by cardiac puncture 90 min following stimulus (CNO or TMT). To minimize stress response in mice, animals were acclimated to the room for a few hours prior to treatment. The same experimenter performed handling during an experiment. Corticosterone levels were quantified in 100 times diluted plasma samples by an ELISA kit (ADI-901-097; Enzo Life Sciences), which was performed according to the manufacturer's guidelines.

## BAT mitochondrial respiration assay

10−12-week-old Tbx21-Cre-/ R26-LSL-hM3Dq-DREADD+ or Tbx21-Cre + / R26-LSL-hM3Dq-DREADD+ were used to measure OCR in BAT upon chemogenetic activation of mitral cell layer of MOB in mice. 90 min after injection of CNO mice were sacrificed and interscapular BAT was collected. Approximately 9 mg of the tissue was cut using a scalpel and placed in a small petri dish containing wash media (Eagle's minimal essential medium (DMEM) (L0102-500, Biowest) supplemented with 25 mM glucose and 25 mM 4-(2-hydroxyethyl)-1-piperazinemethanesulfonic acid (HEPES). The tissue was then placed in an XF24 Islet Capture Microplate (Seahorse Bioscience, North Billerica, MA) with the use of the capture screens and then rinsed twice with wash media, and once with assay media (AM, DMEM with 25 mM glucose). In the end, 450 μl of assay media was added to the tissue and the plate was incubated at 37 °C without $CO_2$ for 45 min. Then, the following final concentrations of the inhibitors were added 24 μg. mL−1 oligomycin, 0.8 μM FCCP, 5 μM rotenone, and 15 μM antimycin A. All the dilutions were freshly prepared. The concentrations of the inhibitors added to each corresponding port in the microplate were ten times higher, to achieve the indicated final concentrations in the assay. Each measurement consisted of 3 min of mixing, 2 min wait time, and 3 min of continuous measuring of $O_2$ levels. OCR was calculated by plotting the $O_2$ tension of the media as a function of time (pmol/min). A more detailed protocol is described in Calderon-Dominguez et al.[60].

## Immunohistochemistry

Transgenic Tbx21-Cre + / R26-LSL-hM3Dq-DREADD+ received saline or CNO (1 mg/kg body weight, i.p.) and were sacrificed 90 min post-injection. Mice were anesthetized with isoflurane and perfused transcardially with ice-cold 0.1 M potassium phosphate-buffered saline (PBS, pH 7.4) followed by 4% PFA. Brains were collected then stored in the same fixatives at 4 °C overnight, then transferred to 30% sucrose at 4 °C overnight. The brains were embedded in optimal cutting temperature (OCT) compound (Tissue-Tek) and cut into 40 microns coronal sections at the region of interest (ROI) on a Leica cryostat. The ROI was confirmed with Paxinos and Franklin's mouse brain atlas and QuickNII tool[61]. For each ROI, four sections were collected anterior to posterior from bregma as follows: Main olfactory bulb: +4.6 mm to

+4.2 mm from bregma; Accessory olfactory bulb: +3.5 mm to +3.1 mm from bregma, Mediobasal hypothalamus (ARC, DMH, VMH and LH), Medial amygdala and Basolateral amygdala: −1.6 mm to −2.0 mm from bregma. Brain sections were incubated in PBS buffered blocking solution containing 2% normal donkey serum and 0.2% Triton X-100 for 1 h at room temperature, followed by incubation overnight at 4 °C in blocking solution containing primary antibodies: anti-NeuN (1:500, PA5-78499, Thermo Fisher Scientific), anti-mCherry (1:500, PA5-34974, Thermo Fisher Scientific), anti-cFos (1:1000, ab190289, Abcam), anti-GFAP (1:1000, 173004, Synaptic Systems). The sections were washed three times in PBS and then incubated in one of the following secondary antibodies for 1 h in the dark at room temperature: donkey anti-rabbit Alexa Fluor® 488 (1:500, A21206, Thermo Fisher Scientific), donkey anti-rabbit Alexa Fluor® 594 (1:500, A21207, Thermo Fisher Scientific), goat anti-guinea pig Alexa Fluor® 488 (1:500, A11073, Thermo Fisher Scientific) and goat anti-rabbit Alexa Fluor® 405 (1:500, A48254, Thermo Fisher Scientific). Sections were washed three times in PBS, mounted onto slides, coverslipped with ProLong® Gold Antifade Reagent with DAPI (8961 S, Cell Signaling). For cFos quantification, ROI sections were stained with a cFos antibody (1:1000, ab190289, Abcam). Fluorescent sections were imaged and z-stacked with a BZ-X700 Keyence. 10X and 20X images were converted to 8-bit grayscale and adjusted for brightness and contrast to best visualize all cFos positive cells. cFos positive cells within ROI were counted blindly using ImageJ v.1.53k (NIH). The exact field of view (FOV) for each region was estimated using BZ-X700 Keyence Analyzer tool (BZX Analyzer v.1.3.1.1.). The mean number of nuclei/mm3 for each mouse was estimated.

## In-situ hybridization combined with immunohistochemistry

The collection and processing of the brain was performed similarly as in the immunohistochemistry method. Brains were cut into 16 microns coronal sections at the ROI on a Leica cryostat. For each ROI, five sections were collected anterior to posterior from bregma as follows: Paraventricular hypothalamus and Central amygdala: −0.6 mm to −1.0 mm from bregma, Mediobasal hypothalamus and Medial amygdala: −1.6 mm to −2.0 mm from bregma. Tissue sections were first fixed in 4% PFA at 4 °C for 15 min and then dehydrated by submerging the slides in graded concentrations of ethanol. In a Coplin jar, slides were first submerged in 50%, then 70% and finally in absolute ethanol, for 5 min each at room temperature. Sections were then treated with hydrogen peroxidase for 5 min at RT followed by submerging the slide rack into the hot 1X co-detection target retrieval (323180, ACD Bio) solution for 5 min. Slides were washed with distilled water and PBS-T. Primary antibody (1:250, ab190289, Abcam) diluted in Co-detection antibody diluent (323160, ACD Bio) were applied, followed by incubation overnight at 4 °C. Next day, slides were washed in PBS-T and fixed in 4% PFA at RT for 30 min. In-situ hybridization was done using RNAscope® multiplex fluorescent reagent kit v2 (323100, ACD Bio), following the manufacturer's protocols. Used RNAscope® probes are as follows: Mm-Slc32a1-C2 (319191-C2, ACD Bio), Mm-slc17a6-C3 (319171-C3, ACD Bio), Mm-LepR (402731, ACD Bio), Mm-Cck (402271, ACD Bio), Mm-Crh (316091, ACD Bio). Post in situ hybridization, slides were washed in PBS-T and secondary donkey anti-rabbit Alexa Fluor® 594 (1:500, A21207, Thermo Fisher Scientific) was applied for 30 min at RT. Sections were washed in PBS-T, mounted onto slides, and coverslipped with ProLong glass antifade mountant (P36980, Invitrogen). Analysis of cells co-expressing cFos and some of the tested genes was performed by QuPath v0.3.2 and v.0.4.0, an open-source platform for image analysis, using positive cell future with a threshold set up determined for each of cFos/genes and kept through all sections.

## Whole-brain tissue clearing and light-sheet imaging

Whole brain vasculature staining was performed following the iDISCO + protocol previously described[62] with minimal modifications. All the

protocol steps were done at room temperature with gentle shaking unless otherwise specified. All the buffers were supplemented with 0,01% Sodium Azide (Sigma-Aldrich, Germany) to prevent bacterial and fungi growth. Perfused brains were dehydrated in an increasing series of methanol (Sigma-Aldrich, France) dilutions in water (washes of 1 h in methanol 20%, 40%, 60%, 80% and 100%). An additional wash of 2 h in methanol 100% was done to remove residual water. Once dehydrated, samples were incubated overnight in a solution containing a 66% dichloromethane (Sigma-Aldrich, Germany) in methanol, and then washed twice in methanol 100% (4 h each wash). Samples were then bleached overnight at 4 °C in methanol containing a 5% of hydrogen peroxide (Sigma-Aldrich). Rehydration was done by incubating the samples in methanol 60%, 40% and 20% (1 h each wash). After methanol pretreatment, samples were washed in PBS twice 15 min, 1 h in PBS containing a 0,2% of Triton X-100 (Sigma-Aldrich), and further permeabilized by a 24 h incubation at 37 °C in Permeabilization Solution, composed by 20% dimethyl sulfoxide (Sigma-Aldrich), 2,3% Glycine (Sigma-Aldrich, USA) in PBS-T. Primary anti-cFos (1:5000, ab190289, Abcam) were incubated for 5 days at 37 °C with gentle shaking, then washed in PBS-T (twice 1 h and then overnight), and finally newly incubated for 4 days with secondary antibody. Secondary antibody conjugated to Alexa 647 (1:5000, ab 150075, Abcam) was used to detect c-Fos. After immunostaining, the samples were washed in PBS-T (twice 1 h and then overnight), dehydrated in a methanol/water increasing concentration series (20%, 40%, 60%, 80%, 100% one hour each and then methanol 100% overnight), followed by a wash in 66% dichloromethane – 33% methanol for 3 h. Methanol was washed out with two final washes in dichloromethane 100% (15 min each) and finally the samples were cleared and stored in dibenzyl ether (Sigma-Aldrich) until light-sheet imaging. Cleared samples were imaged in horizontal orientation (ventral side up) on a light-sheet microscope (Ultramicroscope II, LaVision Biotec). The samples were scanned with a step-size of 5 μm using the continuous light sheet scanning method with the included contrast blending algorithm for the 640 nm and without horizontal scanning for the 480 nm channel. Imaris v.9.6.1 was used for a 3d reconstruction.

## Cell sorting

Hypothalami were dissected from 14 Ai14 mice that were previously injected with AAV.hSyn.Cre.WPRE.hGH (Addgene, 105553-AAV1) bilaterally in each olfactory bulb. Mice were euthanized with $CO_2$ followed by cervical dislocation before brain collection. Dissected tissues from 7 mice were pooled for each sample and cell dissociation was performed using Neural Tissue Dissociation kit (P) (Miltenyi Biotec, Cat#130-092-628) following manufacturer protocol with the use of gentleMACS Octo Dissociator (Miltenyi Biotec, Cat#130-095-937). The cell suspension with added ice-cold HBSS (containing Mg2 + , Ca2 + ) was filtered through a 70 μm cell strainer. The cell suspension was centrifuged at 300 g for 5 min at 4_C. Myelin depletion was performed using Myelin Removal Beads II (Miltenyi Biotec, Cat#130-096-733) using LS columns (Miltenyi Biotec, Cat#130-042-401) according to the manufacturer's instructions. Fluorescent activated sorting was performed on BD FACSAria III sorter with the following settings: nozzle size - 100 mm, pressure - 20 psi. A total of 13742 tdTom+ cells were sorted in both samples and processed by bulk RNA sequencing.

## Bulk RNA sequencing

RNA was isolated using TRIzol/chloroform extraction and RNEasy Qiagen columns. The SMART-Seq V4 Ultra Low RNA Input Kit for Sequencing (Takara Bio USA, Inc., Mountain View, CA) was used for reverse transcription and generation of double stranded cDNA for subsequent library preparation using the Nextera XT Library Preparation kit (Illumina, San Diego, CA). An input of 10 ng RNA was used for oligo(dT)-primed reverse transcription, followed by cDNA amplification and cleanup. Quantification of cDNA was performed using Qubit

(Thermo Fisher Scientific). cDNA normalized to 80 pg/ml was fragmented and sequencing primers added simultaneously. A limiting-cycle PCR added Index 1 (i7) adapters, Index 2 (i5) adapters, and sequences required for cluster formation on the sequencing flow cell. Indexed libraries were pooled and cleaned up, and the pooled library size was verified on the 2100 Bioanalyzer (Agilent Technologies, Santa Clara, CA) and quantified via Qubit. Libraries were sequenced on a NovaSeq 6000 (Illumina) using a 1 × 75 bp read length and coverage of over 30 M reads/sample. Raw reads obtained were aligned to the transcriptome using STAR v.2.6.1 with default parameters, using a custom mouse reference genome (GRCm38.p5).

## Single-cell RNA-sequencing of DMH

Single-cell RNA-seq data from DMH was generated as previously described[63]. Briefly, six 7.5-week-old C57BL/6J mice (3 male, 3 female) were anaesthetized with isoflurane and euthanized. Brains were rapidly extracted and dropped into ice-cold carbogenated NMDG-HEPES-ACSF. Brain sections (2 mm) containing DMH were cut with a razor blade on a stainless-steel brain matrix (51392, Stoelting) and transferred to a dissection dish on ice containing NMDG-HEPES-ACSF. DMH was bilaterally microdissected in -700 × 700 × 350 μm tissue block. The tissue was aggregated in a collection tube on ice containing NMDG-HEPES-ACSF and 30 μM actinomycin D. For enzymatic tissue digestion, NMDG-HEPES-ACSF was replaced by trehalose-HEPES-ACSF containing papain (50 U/ml; P3125, Sigma-Aldrich, pre-activated with 2.5 mM cysteine and a 30 min incubation at 34 °C and supplemented with 0.5 mM EDTA) and 15 μM actinomycin D. Extracted DMH tissue was incubated at room temperature with gentle carbogenation for 55 min. During enzymatic digestion the tissue was pipetted periodically every 10 min. At the end of enzymatic digestion, the medium was replaced with 200 μl of room temperature trehalose-HEPES-ACSF containing 3 mg/ml ovomucoid inhibitor (OI-BSA,Worthington) 25 U/ml DNase I (90083, Thermo Scientific) and 15 μM actinomycin D and tissue was gently triturated into a uniform single-cell suspension with fire-polished glass Pasteur pipettes. The resulting cell suspension volume was brought up to 1 ml with trehalose-HEPES-ACSF with 3 mg/ml ovomucoid inhibitor and pipetted through a 40 μm cell strainer. Single-cell suspension was centrifuged down at 300 g for 5 min at 4 °C and the supernatant was replaced with 1 ml fresh ice-cold trehalose-HEPES-ACSF and the cell pellet was resuspended. Cells were pelleted again and resuspended in 100 μl of ice-cold resuspension-ACSF. Cell suspensions were kept on ice while cell densities were quantified with a hemocytometer and the final cell densities were verified to be in the range of 300–1000 cells/μl. Cell suspension volumes estimated to retrieve -10,000 single-cell transcriptomes were added to the 10x Genomics RT reaction mix and loaded to the 10X Next GEM Chip G (2000177, 10× Genomics) per the manufacturer's protocol. We used the Chromium Next GEM Single Cell 3′ Reagents Kit v3.1 (1000128, 10x Genomics) and the Single Index Kit Set A (100213) to prepare Illumina sequencing libraries downstream of reverse transcription following the manufacturer's protocol. Resulting scRNA-seq sequencing library was sequenced on an Illumina NovaSeq 6000 sequencers (paired-end 150).

Sequencing data were aligned to optimized pre-mRNA reference transcriptome[64] and digital gene-cell matrices were generated with the 10× Genomics Cell Ranger v.6.0.0 count pipeline. The resulting scRNA-seq data were analyzed in R (4.1.2) using Seurat (v.4.1.0.9007) as previously described[63]. Briefly, expression data were filtered to exclude cells with fewer than 1000 unique transcripts as well as cells exhibiting more than 15% of mitochondrial transcripts. Female-specific marker Xist was used to determine sex origin of cells and validated by additional query of male-specific genes Ddx3y, Eif2s3y, Gm29650, Kdm5d, and Uty. We followed the standard Seurat workflow to identify transcriptomic cell-types and assigned cell class identities based on canonical cell-type markers: Ndrg4, Snap25, Eno2 and Tubb3 for

neurons, Ntsr2, Gfap, Fabp7 for astrocytes, Slco1c1, Fn1, Slco1a4 and Cldn5 for endothelial cells, Fcer1g, Aif1 and C1gc for microglia and Mag, Mbp, Ugt8 and Mog for oligodendrocytes. We digitally extracted neuronal clusters from our dataset resulting in expression data for 10028 neurons that we used to establish an unbiased neuronal nomenclature for DMH. From these neurons, 5664 inhibitory and 3972 excitatory neurons were identified via markers Slc32a1 and Slc17a6, respectively. From these, cells were further subclustered and filtered based on expression of either Slc32a1 or Slc17a6. Final clusters were analyzed by Seurat::FindMarkers (v.4.1.0.9007) to identify top genes and verify DMH-specific expression via the in situ::Allen Brain Atlas. For cluster nomenclature, we used an approach that involved first labeling neurons as GABAergic or Glutamatergic subtypes. We then added two of the genes from the list of the 25 most variable genes for that specific cluster. When choosing which two genes to include in the cluster name, we looked for genes that were already functionally associated with DMH, whenever possible.

## Mouse estrus cycling staging identification

The collecting of vaginal cells was performed by vaginal lavage. The collected fluid was placed on a glass slide and allowed the smear to completely dry at room temperature. Estrous smears were stained using crystal violet. Staining was examined by microscopy to determine cell types present. The ratio of cells present (cornified squamous epithelial cells, leukocytes, and/or nucleated epithelial cells) was used to determine the estrous stage of mouse at the time of sample collection.

## Statistical analysis

GraphPad Prism v.8.4.3 was used for statistical analysis. Data are expressed as mean ± SEM. $p$ Values were calculated using t-student test, one-way, two-way ANOVA, three-way ANOVA or mixed model followed with an appropriate post-hoc tests. Analysis of covariance (ANCOVA) was performed in R. $p$ values < 0.05 were considered significant. Results of statistical tests can be found in Supplementary tables S1–11.

## Reporting summary

Further information on research design is available in the Nature Portfolio Reporting Summary linked to this article.

## Data availability

All data generated and analyzed during this study are included in this paper and its supplementary information files. All sc-RNA seq data of mouse dorsomedial hypothalamus that are used in this study have been deposited in the the National Center for Biotechnology Information Gene Expression Omnibus (GEO) and are accessible through the GEO Series accession number: GSE232230. Mouse reference transcriptome v.2 used in scRNA-seq analysis is available for download at www.thepoollab.org/resources and custom mouse reference genome GRCm38.p5 used for bulk RNA-seq can be found under RefSeq assembly accession: GCF_000001635.25. Source data are provided with this paper.

## Code availability

The R code used to perform the scRNA-seq analysis is available from the corresponding author on request.

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

## Acknowledgements

We thank the Applied Genomics core at CSMC for their assistance in processing RNA-seq, the Biobehavioral core for assisting with open field testing and the Flow Cytometry core for cell sorting. We thank Joshua Breunig for his assistance with light-sheet imaging, Michael Ramos and Ritchie Ho for their advice toward scRNA analysis. This work was supported by the American Diabetes Association Pathway to Stop Diabetes Grant 1-15-INI-12 (C.E.R.), the Klingenstein-Simons foundation (C.E.R.), the Larry L Hillblom Foundation fellowship 2019-D-014-FEL (P.J.) and the Cedars-Sinai Center for Research in Women's health and Sex Differences pilot award (C.E.R). Schematic cartoons were created with BioRender.com.

## Author contributions

C.E.R. and P.J. conceived the studies. P.J. performed experiments with assistance from Y.W., E.N., N.K. and K.J. A.H.P. and Y.O. executed the scRNAseq experiments, and bioinformatics analysis was conducted by P.J. N.M. and A.H.P. C.E.R. and P.J. wrote the manuscript with assistance from Y.O. and A.H.P. C.E.R. supervised the entire study.

## Competing interests

The authors declare no competing interests.
