## [Peer Review File · Nature Communications]

A Sex-Specific Thermogenic Neurocircuit Induced By Predator Smell Recruiting Cholecystokinin Neurons In The Dorsomedial HypothalamusREVIEWER COMMENTS

Reviewer #1 (Remarks to the Author):

In the current contribution Jovanovic et al., investigate the sex-specific effects of predator odorants on metabolism in mice. They demonstrate that TMT elicits a predominant increase on O₂ consumption in female but not male animals. This is accompanied by a comparable increase of TMT-elicited Fos immunoreactivity in the ARC and DMH of mice of both genders. Chemogenetic activation of TbxCre-expressing neurons in the MOB elicits a similar degree of Fos activation in the MOB of male and female animals, however, only female mice respond with an increase in O₂ consumption and BAT temperature. This effect persists upon ovariectomy of female mice. These effects are accompanied by comparable CNO-induced Fos expression in the ARC and LH, yet a more pronounced cell activation in the DMH of female mice. Through polysynaptic anterograde tracing from the MOB, they identified cells in the LH and DMH receiving polysynaptic inputs from the MOB, and single cell sequencing revealed innervation of CCK-expressing neurons in the DMH among others. Chemogenetic activation of DMH CCK neurons resulted in increased energy expenditure.

In summary, this study reveals interesting new findings, which in principle should be of interest to the audience of Nature Communications. However, a few major points should be addressed to support the conclusions drawn.

1. The study largely centers around the sex difference in responses. However, it remains unclear at what level of the circuitry this is encoded. TMT elicits similar Fos reactivity in the DMH of male and female mice, yet the physiological response (O₂ consumption) is different. Thus, the direct gender-specific activation of CCK DMH neurons in response to TMT should be tested.
2. Similarly, the effect of chemogenetic MOB neuron activation on CCK Fos expression in CCK neurons should be quantified in both genders.
3. The effect of chemogenetic DMH CCK neuron activation should be assessed in male mice.
4. Finally the authors state that „CCK neurons mediate the increased energy expenditure caused by olfactory neurons' activation“ without actually proving this. They rather show that CCK DMH neuron activation can lead to increased energy expenditure, without showing

their necessity for odorant-induced energy expenditure. The proper experiment would be to chemogenetically inhibit CCK DMH neurons and to address whether this blocks TMT-induced increases in energy expenditure.

Reviewer #2 (Remarks to the Author):

In this manuscript the authors explore the role of olfactory neuron activation (both with chemogenetics and a predator odor) in energy metabolism, which is a very interesting and somewhat underexplored topic. One of the most striking observations is that olfactory neuron activation increases energy expenditure (likely through brown adipose tissue activation) without increasing physical activity in the case of chemogenetic activation. This is a striking observation, and the observation that DMH CCK neuron activation increases oxygen consumption is too. The indirect calorimetry data is clear but can possibly be improved. Similarly, the neuroanatomy of the DMH needs some clarifications to make it up to the standard. Furthermore, several experimental details are missing and typos need to be corrected.

Major issues:

1) Fig 1c,d,e,j,i: The results section mentions that the male mice “switched to freezing behavior” and the females also during the dark phase. Has that been quantified or observed with video recordings? Freezing is a specific behavior that is not usually inferred from reduced physical activity levels only. The physical activity graph shows physical activity levels similar to before the swab presentation. So, in the absence of direct evidence of freezing behavior, this may be a speculation and not a conclusion.

2) Fig 1 o-s, Could you please clarify the following aspects?

- In how many slices per mouse was each structure counted? And at which bregma coordinates? Eg for the DMH, 3 sections per mouse between br -1.5 and -2.3 were counted
- Usually, the abbreviation cDMH refers to the compact portion of the DMH, which runs diagonally from ventromedial to dorsolateral in part of the DMH. And what do you refer to as the caudal DMH?
- It is mentioned that the “cFos content within dDMH and cDMH was more abundant in females compared to males”. This is clear from the image in Fig 1, but has this been

quantified? Again, what exactly are the DMH subregions that are being referred to?

3) In the Tbx21-Cre:hM3Dq experiment, how come that after the Veh/CNO administration the physical activity decreases (from 0 to 45 minutes), but the VO₂ does not change (Eg fig 2g and k)? This is surprising because VO₂ consumption virtually always correlates with physical activity, as you can see clearly in all groups in fig 1c,d and g,h. Would it be possible to graph some period (perhaps 60 min) of the VO₂ and physical activity before Veh/CNO administration? You do this beautifully in Fig 1. Alternatively, what could be an explanation for not seeing an increase in VO₂?

4) Fig 4, similar to comment 2) above, could you please specify what bregma levels and how many “slices” or images were counted per structure? And at what bregma levels?

5) Fig 5a: some brain region abbreviations are missing. It is further a little unclear what exactly is happening here. Generally, the virus used for this experiment (AAV-hSyn-DIO-mCherry (Addgene, 50459-AAV8)) is not thought to anterogradely infect neurons, unlike some AAV1 viruses, as used in f5b. Therefore, what you would expect is anterograde projections of MOB neurons in the areas that MOB neurons are synaptically connected to. But the text states: “Mapping out connections of OB Tbx21 neurons by anterograde tracing using the presynaptic tracer AAVhSyn- DIO-mCherry in MOB of Tbx21-Cre mice labelled mitral/tufted cells in the MOB, the lateral olfactory tract (LOT), olfactory cortex (Piriform, PIR) and olfactory tubercula (OT)” and there seem to be cell bodies visible in eg the PIR in fig 5a. Also, LOT in fig 5a is oversaturated.

6) Fig 5d,e “Fluorescence activated cell sorting (FACS) of TdTomato-expressing cells arising from the microdissected hypothalamus revealed enrichment of transcripts” should possibly be something like: FACS followed by sequencing of TdTom cells. Also, apologies if I missed it, but I did not see any information on the bulk-sequencing/enrichment in the methods, other than a description of the FACS process. For instance, how many TdTom+ cells were used? Also, what could be a reason that CCK is not enriched in the TdTom cells?

7) Single cell sequencing experiment: As best as I know, yours would be the first published DMH scRNAseq data set. As a reference, therefore, it would be rather useful to include a table such as Fig S4 and S5 in Moffit, 2018 (PMID: 30385464) and/or an expression profile figure/heatmap with several markers for each cluster. As such, fig S8c-e from the snRNAseq experiment is helpful, but it could be slightly more accessible: split by excitatory vs inhibitory cluster, and add some markers for each cluster.

8) Single nucleus sequencing experiment: if the LH was included, why is there no mention of typical LH markers? Also, GAD2 may be a promiscuous marker for GABAergic neurons Moffit, 2018 (PMID: 30385464). In Fig 5i (top left), there seem to be several CCK-expressing clusters. Were those not detected in the snRNAseq experiment (Fig S8e)? Out of interest, why did you choose to perform a snRNAseq experiment to have “confirmed that Cck-expressing neurons were present in independent cell types non-overlapping with LepR+ neurons” and not use the scRNAseq dataset? Also is there a way to quantitatively compare (or combine?) these two datasets? This would enhance any drawn conclusions.

9) CCK neurons in the DMH. As Imoto et al., 2021 describe, the DMH CCK neurons are specifically located in the compact portion of the DMH (cDMH). You can sort of see this too in the left DMH in Fig 5c as the diagonal band running from ventromedial to rostralateral. This will become even clearer in the slices more caudal to the one chosen to show in fig 6c. Could you find one more caudal that reflects this specific pattern of CCK being a marker for the cDMH better? Similarly, if you would zoom out a little in fig 6a, one would be able to appreciate this aspect. The fig 6a panels are rather large, so maybe a zoomed out image with the cDMH visible and an inset with the FOS detail could be a good option. Furthermore, you mention that “Cck-expressing neurons were predominantly in the dDMH region,” which does not correspond with the literature and the Allen brain atlas (see image 25 and 26 of the coronal CCK sections <http://mouse.brain-map.org/experiment/siv?id=77869074&imageId=77809652&initImage=ish&coordSystem=pixel&x=6060m&y=5668&z=3>). Could you please address this? Do CCK neurons overlap with the DMH BRS3 population? Stimulation of BRS3 DMH neurons also increases EE (Pinol, 2018; PMID: 30349101).

10) Since the implication of DMH CCK neurons in the MOB->PIR/OT/other?->DMH pathway is indirect (“Ttr clusters were abundant in Cck” and FOS staining in CCK in neurons), it could be worthwhile to increase evidence for this pathway or maybe somewhat attenuate the statements/conclusions that the DMH CCK neurons mediate the MOB activation/TMT induced VO2 increase. Ideally one would a) inhibit/silence/kill the CCK neurons in an experiment Tbx21-Cre:hM3Dq experiment, or b) show with monosynaptic retrograde rabies tracing from CCK neurons and CRACM that the PIR/OT/other? neurons that project to DMH CCK neurons receive direct synaptic input from MOB neurons.

Minor issues:

- This (p5) sounds a little garbled: “the activatory Gq DREADD receptor cholinergic receptor muscarinic 3”
- Please specify if the Tbx21 Cre- controls are littermates of the Cre+ mice used in the experiment
- Fig2o-q It is somewhat hard to measure Tbat without shaving the back of the mouse because the hair insulates. That is why the temperature is so low.
- Sf2a, f4a and some other figs miss scale bar
- “declined after 3 hours and returned to baseline levels (Supp fig. 4d)” has to be S2d
- Regarding the AAV-DIO-hM3Dq in Tbx21-Cre mice approach: “larger number of mitral cells are required”. Could it be that the mitral cells that can drive EE are located in another part of the MOB?
- Sf2i: the increase in OCR in females is higher than in males, regardless of the genotype. What statistical test was used here? Also, is this an established phenomenon?
- “considered that Cre+ female’s specific” is probably females’
- Fig 3h. Please add somewhere what the period of post CNO is?
- “The MeA has been heavily involved”; maybe “is” instead of “has been”?
- Fig4e. Since you are later focusing on the compact DMH, would it be possible to quantify the colocalization of Vglut2 and vgat with FOS in the subregions of the DMH (dorsal DMH, compact DMH, ventral DMH – and if you specify what exact region you refer to as caudal DMH)?
- “MOB of Ail14 (LSL-TdTomato reporter) mice” (p7) should be Ai14
- “in selected postsynaptic neuronal targets, thus allowing axonal tracing” should possibly be: “selectively in postsynaptic neuronal targets, allowing polysynaptic anterograde tracing”
- “identified 41 neuronal clusters (Fig. 5h)” should be 5g.
- “Our results suggest so far indicate that DMH neurons” is garbled
- Was the FOS expression in fig 6a and fig s7c quantified?
- I may have overlooked it, but is it mentioned anywhere if both uni and bilaterally expressing hMDdq mice were included in the analyses in the Slc1a3-Cre and CCK-Cre groups? I read that you injected the virus bilaterally, but that does not always translate in bilateral expression, irrespective of your brilliant stereotactic surgery skills.
- The discussion says: “We therefore propose the neurocircuit MOB→VgatMeA→ DMH

CCK” but this seems contradictory to some results in the manuscript. For instance you find that DMH CCK neurons are activated, not inhibited. Or are the MOB neurons also inhibitory?

- The last sentence of the discussion is a bit abrupt, since that is the first time in the manuscript that obesity is mentioned.

Reviewer #3 (Remarks to the Author):

The manuscript by Jovanovic and colleagues presents intriguing evidence of a neural circuit involved in a female-specific response to a predator-related odor. Using a red fox predator scent, TMT, they show that female and male mice have divergent responses with respect to energy balance. Specifically, VO₂ consumption and activity are acutely elevated after exposure to TMT and then lowered during the following night. Feeding after an overnight fast is also suppressed by TMT in female mice. In contrast, VO₂ is acutely decreased in males and the effect on feeding after a fast is not as strong as in females.

The subsequent analyses are designed to identify brain regions, cells, and circuits that are involved, as measured by cFOS immunoreactivity, tracing, and transcriptomics. cFOS immunoreactivity is induced in the ARC and DMH, rather selectively by predator odor compared to other odors, but the effects are similar in males and females. FACS RNA-seq and snRNA-seq identify markers of populations within the DMH that are synaptically connected to TBX21+ olfactory bulb neurons. Finally, chemogenetic activation of TBX21-lineage and Cck-expressing neurons partially phenocopy the effect of TMT. Specifically, activating neurons of the TBX21+ lineage with a transgenic hM3Dq recapitulates some of the sex-specific phenotypes and activating Cck-expressing neurons using AAV-hM3Dq recapitulates some of the phenotypes in females, suggesting that activating a circuit that includes TBX21 OB neurons or Cck-expressing neurons of the DMH can partially recapitulate the effect of TMT. The authors also show that ovariectomy does not alter the effect activating neurons of the TBX21+ lineage, consistent with the conclusions that either the effects are not dependent on ovarian hormones or that chemogenetic neuronal activation can bypass any possible effects of ovarian hormones on the circuit.

I think that there are multiple major concerns that should be addressed prior to publication.

Major comments:

Ovariectomy did not block the effect of OB-Tbx21 activation on VO₂. This is interpreted to “indicate that this metabolic response is unlikely to be dependent on estrogens tuning of olfactory performance, but rather might be the consequence of sexually dimorphic neurocircuits”. Although this is one possibility, it is also possible that neuron activation could bypass the need for or modulatory effect of gonads. For example, Mc4r-expressing neurons in the VMHvl regulate estrogen-dependent movement in females, but activating the neurons increases movement in both sexes and in females without ovaries (PMID: 34646010). If the goal is to test the effect of gonads, it would be necessary to do the OVX experiment in the context of TMT odor presentation to ask if ovaries are required for the effects in females. Further, it would be important to include both sexes, with and without gonads, because it is possible that the effect is suppressed by testicular hormones in males. As presented, the OVX manipulation is difficult to interpret.

The conclusion that “a neuronal population in the DMH [is] required for this response” is not supported by the data. Activating Cck+ DMH neurons can partially recapitulate the response in females but the phenotyping is more limited and the experiment does not include males. Even if the phenocopy were perfect, it would be more appropriate to conclude that the neuronal population in the DMH is sufficient for this response. If the goal is to test requirement, the authors would have to show that silencing or ablating Cck+ DMH neurons blocks the response to TMT.

Any claim of a sex difference requires sex (and interactions with sex) to be factors in an ANOVA or mixed model. The methods state that these statistical analyses were performed but I could not find the results of those tests.

Why would females, but not males, increase energy expenditure upon being exposed to predator odor? Is it related to escaping the odor by running away?

To what extent is thermogenesis involved in the response to TMT? Thermogenesis is highlighted in the title but is only evaluated in the context of activating TBX21+ OB neurons.

Is thermogenesis elevated by TMT? Is this a major effect of TMT or major output of the circuit? Why would TMT engage a thermogenic circuit?

Minor Comments

The link to stress is highlighted in the title but is underdeveloped. There is a rise in cort levels but the response is not generalized to other stressful contexts. Perhaps the title and interpretations can be limited to the context of predator-related odor?

Related to cort, the methods do not describe how the blood was collected for measurements in plasma. It would be important to describe how acute stress was avoided in the collection process to avoid inducing a stress response in all mice, including controls, which could mask an effect of TMT in males.

The design of the chemogenetic experiments include saline injection but I cannot find the phenotyping of mice after saline injection. How does this compare to the effects of CNO in within-subject comparisons?

In the final figure, there are representative images of Cck transcript and cFOS immunoreactivity. It should be specified how many mice were imaged and the images should be quantified.

Dear Colleagues,

We wish to thank you for your comments and critiques of our study. As you will find, we have carefully addressed each comment and suggestions and have summarized them below. We have addressed the points raised by the reviewers. We have added additional experiments, including chemogenetic activation of DMH-CCK neurons in both males and females, as well as silencing of these neurons in the presence of predator odor exposure. We also have thoroughly explored the role of DMH-CCK neurons in sensing predator odor and DREADD stimulation of olfactory neurons using quantification of fluorescent in situ hybridization assays, and added additional results examining the recruitment of stress sensing CRH neurons in the amygdala and paraventricular nucleus of the hypothalamus. We have further analyzed our single-cell RNA-sequencing results to better characterize dorsomedial hypothalamic neurons populations, as our study is the first one to date to provide a neuronal atlas of this region. We also report higher Cck expression in females compared to males in DMH neurons, which coincides for enhanced thermogenic output upon chemogenetic activation of these neurons. Finally, our silencing experiment of DMH-CCK neurons implicates these neurons in predator odor-induced thermogenesis. We hope that you will agree that these results improve the quality of the manuscript. We thank you for considering this body of work for publication in Nature Communications.

Kind regards,
Celine Riera

REVIEWER COMMENTS

Reviewer #1 (Remarks to the Author):

In the current contribution Jovanovic et al., investigate the sex-specific effects of predator odorants on metabolism in mice. They demonstrate that TMT elicits a predominant increase on O2 consumption in female but not male animals. This is accompanied by a comparable increase of TMT-elicited Fos immunoreactivity in the ARC and DMH of mice of both genders. Chemogenetic activation of TbxCre-expressing neurons in the MOB elicits a similar degree of Fos activation in the MOB of male and female animals, however, only female mice respond with an increase in O2 consumption and BAT temperature. This effect persists upon ovariectomy of female mice. These effects are accompanied by comparable CNO-induced Fos expression in the ARC and LH, yet a more pronounced cell activation in the DMH of female mice. Through polysynaptic anterograde tracing from the MOB, they identified cells in the LH and DMH receiving polysynaptic inputs from the MOB, and single cell sequencing revealed innervation of CCK-expressing neurons in the DMH among others. Chemogenetic activation of DMH CCK neurons resulted in increased energy expenditure.

In summary, this study reveals interesting new findings, which in principle should be of interest to the audience of Nature Communications. However, a few major points should be addressed to support the conclusions drawn.

1. The study largely centers around the sex difference in responses. However, it remains unclear at what level of the circuitry this is encoded. TMT elicits similar Fos reactivity in the DMH of male and female mice, yet the physiological response (O2 consumption) is different. Thus, the direct gender-specific activation of CCK DMH neurons in response to TMT should be tested.

We thank the reviewer for this excellent point. We have added additional experiments to answer this specific question. In particular, we have tested the sex-specific recruitment of the compact and dorsal DMH-CCK neurons by these stimuli. We found that compact DMH-CCK neurons are activated by chemogenetic stimulation in females, and we observe a similar trend upon TMT exposure (Fig. 7). We also tested the effects of stimulating DMH-CCK neurons with chemogenetics in both sexes. We find that in both males and females, activation of these neurons results in enhanced energy expenditure, yet the effects appear greater in females (Fig. 8c,d). An interesting observation arising from the single cell RNA-seq data is that Cck expression in these neurons is higher in females compared to males (Fig. 6l). Taken together, these data suggest that the processing of stimuli in a sex-dependent way is likely to occur upstream of DMH-CCK neurons, such as in the medial amygdala where we find a female specific recruitment of VGAT neurons.

2. Similarly, the effect of chemogenetic MOB neuron activation on CCK Fos expression in CCK neurons should be quantified in both genders.

This now has been added (Fig. 7a,b). We find increased recruitment of compact DMH-CCK neurons in females compared to males upon MOB chemogenetic stimulation.

3. The effect of chemogenetic DMH CCK neuron activation should be assessed in male mice.

This is now shown in Fig. 8 c,d.

4. Finally the authors state that "CCK neurons mediate the increased energy expenditure caused by olfactory

*neurons' activation "without actually proving this. They rather show that CCK DMH neuron activation can lead to increased energy expenditure, without showing their necessity for odorant-induced energy expenditure. The proper experiment would be to **chemogenetically inhibit CCK DMH neurons** and to address whether this blocks TMT-induced increases in energy expenditure.*

We agree with the reviewer. Instead of chemogenetic inhibition, we have opted for tetanus toxin neuronal silencing in order to avoid having to inject mice with CNO prior to TMT odor testing and limit potential confounding of stress handling. These new results are now shown in Fig. 8 h-m. We have found that silencing DMH-CCK neurons reduced the amplitude of the energy expenditure response to TMT. Importantly, there remains a stress-induced heat production in DMH-CCK silenced mice which suggests that other neuronal chemotypes might contribute to this response (such as PVN-CRH neurons).

Quite interestingly, silencing DMH-CCK neurons did not alter the activity-induced burst following TMT suggesting that other neurons are involved in that behavioral response to stress. Rather, silencing DMH-CCK neurons increased feeding following TMT exposure as well as no odor, demonstrating an important role for these neurons in feeding.

Reviewer #2 (Remarks to the Author):

In this manuscript the authors explore the role of olfactory neuron activation (both with chemogenetics and a predator odor) in energy metabolism, which is a very interesting and somewhat underexplored topic. One of the most striking observations is that olfactory neuron activation increases energy expenditure (likely through brown adipose tissue activation) without increasing physical activity in the case of chemogenetic activation. This is a striking observation, and the observation that DMH CCK neuron activation increases oxygen consumption is too. The indirect calorimetry data is clear but can possibly be improved. Similarly, the neuroanatomy of the DMH needs some clarifications to make it up to the standard. Furthermore, several experimental details are missing and typos need to be corrected.

Major issues:

1) Fig 1c,d,e,j,i: The results section mentions that the male mice "switched to freezing behavior" and the females also during the dark phase. Has that been quantified or observed with video recordings? Freezing is a specific behavior that is not usually inferred from reduced physical activity levels only. The physical activity graph shows physical activity levels similar to before the swab presentation. So, in the absence of direct evidence of freezing behavior, this may be a speculation and not a conclusion.

Thank you for pointing this out, this is a valid concern. As we have not conducted video recordings to characterize the observed behavior, we have removed this sentence from the results section and focused on describing the immediate phenotypes associated with TMT presentation, which is elevated locomotor activity.

2) Fig 1 o-s, Could you please clarify the following aspects?

- In how many slices per mouse was each structure counted? And at which bregma coordinates? Eg for the DMH, 3 sections per mouse between br -1.5 and -2.3 were counted.*

We now have updated the methods section to clearly explain the experimental procedures and associated quantification.

"For each ROI four sections were collected anterior to posterior from bregma as follows: Main olfactory bulb: +4.6mm to +4.2mm from bregma; Accessory olfactory bulb: +3.5mm to +3.1mm from bregma, Mediobasal hypothalamus (ARC, DMH, VMH and LH), Medial amygdala and Basolateral Amygdala: -1.6mm to -2.0mm from bregma"

- *Usually, the abbreviation cDMH refers to the compact portion of the DMH, which runs diagonally from ventromedial to dorsolateral in part of the DMH. And what do you refer to as the caudal DMH?*

This is was an error in description, we thank the reviewer for bringing this up. We also meant compact.

- *It is mentioned that the "cFos content within dDMH and cDMH was more abundant in females compared to males". This is clear from the image in Fig 1, but has this been quantified? Again, what exactly are the DMH subregions that are being referred to?*

We agree that we reported a visual observation without the support of any quantification (dDMH and cDMH). The way we quantified was to collect 5 sections per well and selected 4 randomly for quantification. Therefore, we have not obtained that level of information in Fig 1. We have now removed this statement from the result and included at a later stage in the manuscript. We have thoroughly investigated the presence of CCK neurons in the various areas of the DMH, and found that most CCK neurons were localized in the compact region, where we observed a female-specific activation of these neurons (Fig. 7). This was quantified for both MOB chemogenetic stimulation and TMT exposure in both sexes.

3) In the Tbx21-Cre:hM3Dq experiment, how come that after the Veh/CNO administration the physical activity decreases (from 0 to 45 minutes), but the VO₂ does not change (Eg fig 2g and k)? This is surprising because VO₂ consumption virtually always correlates with physical activity, as you can see clearly in all groups in fig 1c,d and g,h. Would it be possible to graph some period (perhaps 60 min) of the VO₂ and physical activity before Veh/CNO administration? You do this beautifully in Fig 1. Alternatively, what could be an explanation for not seeing an increase in VO₂?

Unfortunately, the way we performed the chemogenetic experiment differed from the odor test. In the odor test, we briefly open the metabolic cage (around 2s) to introduce the stimulus. There is no loss of data or massive re-equilibration of the gas in the cage. The mouse is intrigued and comes to explore; therefore, activity goes up.

However, in the chemogenetic assay, the animal is extracted from the cage and injected ip, which takes a couple of minutes. Each animal is injected at a slightly different time. For a total of 8 metabolic cages, it takes about 5-10 minutes to complete the injections. This requires to re-launch the program and changes the VO₂ baseline. We therefore don't have a continuous monitoring of VO₂ before injection. We agree that this is a limitation of this methodology. Our supplementary Fig. 2b shows additional controls demonstrating that Cre⁺ and Cre⁻ have similar baseline VO₂.

Regarding the locomotor activity, as the program starts approximately about 5 min post injections, the animals are stressed from the ip injections and are very active at first, before calming down within the following minutes.

The different experimental approaches to induce stress used in Figs.1 and 2 led us to present the results in this fashion.

4) Fig 4, similar to comment 2) above, could you please specify what bregma levels and how many "slices" or images were counted per structure? And at what bregma levels?

Yes, we now included this statement in the methods. "The brains were embedded in optimal cutting temperature (OCT) compound (Tissue-Tek) and cut into 40 microns coronal sections at the region of interest (ROI) on a Leica cryostat. The ROI was confirmed with Paxinos and Franklin's mouse brain atlas and QuickNII tool⁵⁹. For each ROI four sections were collected anterior to posterior from bregma as follows: Main olfactory bulb: +4.6mm to +4.2mm from bregma; Accessory olfactory bulb: +3.5mm to +3.1mm from bregma, Mediobasal hypothalamus (ARC, DMH, VMH and LH), Medial amygdala and Basolateral Amygdala: -1.6mm to -2.0mm from bregma"

5) Fig 5a: some brain region abbreviations are missing. It is further a little unclear what exactly is happening here. Generally, the virus used for this experiment (AAV-hSyn-DIO-mCherry (Addgene, 50459-AAV8)) is not thought to anterogradely infect neurons, unlike some AAV1 viruses, as used in f5b. Therefore, what you would expect is anterograde projections of MOB neurons in the areas that MOB neurons are synaptically connected to. But the text states: "Mapping out connections of OB^{Tbx21} neurons by anterograde tracing using the presynaptic tracer AAVhSyn- DIO-mCherry in MOB of Tbx21-Cre mice labelled mitral/tufted cells in the MOB, the lateral olfactory tract (LOT), olfactory cortex (Piriform, PIR) and olfactory tubercula (OT)" and there seem to be cell bodies visible in eg the PIR in fig 5a. Also, LOT in fig 5a is oversaturated.

Thank you for this comment, we have modified the text to reflect this detail.

"To characterize the neurons responsible for stress-induced hyperthermia, we mapped out connections of OB^{Tbx21} neurons by anterograde tracing using two different viral strategies. First, we assessed anterograde projections of OB^{Tbx21} neurons in the synaptically connected regions using the tracer AAV-hSyn-DIO-mCherry delivered to the MOB of Tbx21-Cre mice. We observed mCherry signal in mitral/tufted cells, the lateral olfactory tract (LOT), olfactory cortex (Piriform, PIR) and olfactory tubercula (OT) but there was no direct projections to the MBH (Fig 6a)."

6) Fig 5d,e "Fluorescence activated cell sorting (FACS) of TdTomato-expressing cells arising from the microdissected hypothalamus revealed enrichment of transcripts" should possibly be something like: FACS followed by sequencing of TdTom cells. Also, apologies if I missed it, but I did not see any information on the bulk-sequencing/enrichment in the methods, other than a description of the FACS process. For instance, how many TdTom+ cells were used? Also, what could be a reason that Cck is not enriched in the TdTom cells?

Thank you for this suggestion. We modified the sentence related to Figure 5d,e : "Fluorescence-activated cell sorting (FACS) followed by sequencing of TdTomato-expressing cells from the microdissected hypothalamus revealed enrichment of transcripts including Ttr, Vim, Nnat, Ntsr2 and Clu (Fig. 6d, e)". We also incorporated a more detailed bulk RNA-sequencing protocol into the *Material and methods* section and the total number of tdTom cells. Data from tdTom+ bulk RNA-seq revealed the presence of Cck gene below our threshold of the top 500 genes. A possibility as to why we failed to detect this transcript in bulk RNA-seq was our inclusion of the entire dissociated hypothalamus in the FACS analysis which has affected the enrichment of this small neuronal population. Additionally, we have observed in our in-situ hybridization assays that Cck expression is less abundant

in neurons of the DMH over other brain regions containing Cck neurons such as the amygdala, where it is known to be strongly expressed.

7) Single cell sequencing experiment: As best as I know, yours would be the first published DMH scRNAseq data set. As a reference, therefore, it would be rather useful to include a table such as Fig S4 and S5 in Moffit, 2018 (PMID: 30385464) and/or an expression profile figure/heatmap with several markers for each cluster. As such, fig S8c-e from the snRNAseq experiment is helpful, but it could be slightly more accessible: split by excitatory vs inhibitory cluster, and add some markers for each cluster.

We are thankful to the reviewer for pointing out this. This would, indeed, be, to our knowledge, the first published DMH scRNA-seq data set. We agreed with the reviewer's comments and included a more detailed analysis of DMH neuronal population in Fig. 6 i,j where we presented 30 inhibitory clusters (GABA1-GABA-30) and 26 excitatory clusters (GLUT1-GLUT26) with nomenclature. We also showed UMAPs of inhibitory and excitatory clusters with nomenclature and graphs representing the list of a top genes in each cluster in Supplementary Fig 7.

8) Single nucleus sequencing experiment: if the LH was included, why is there no mention of typical LH markers? Also, GAD2 may be a promiscuous marker for GABAergic neurons Moffit, 2018 (PMID: 30385464). In Fig 5i (top left), there seem to be several CCK-expressing clusters. Were those not detected in the snRNAseq experiment (Fig S8e)? Out of interest, why did you choose to perform a snRNAseq experiment to have "confirmed that Cck-expressing neurons were present in independent cell types non-overlapping with LepR+ neurons" and not use the scRNAseq dataset? Also is there a way to quantitatively compare (or combine?) these two datasets? This would enhance any drawn conclusions.

We have performed more thorough analysis of our scRNA-seq data of the DMH, and acknowledge that several neuronal clusters express Cck (Fig. 6i,j). We were able to show that Lepr is present in the scRNAseq data, as well as Brs3 neurons, and that these neurons constitute separate clusters.

We agree that the presentation of both scRNAseq and snRNAseq was confusing for the purpose of this study. In general, although we have found that the snRNAseq dataset confirmed findings obtained from the scRNAseq, multiple cell populations, such as Cck and Lepr, were less well represented, and some others were missing. Cck-expressing neurons in DMH are mainly expressed in dorsal and compact part of the DMH, as we showed in Fig 7. Their high specificity of expression within DMH and not LH, might be one of the reasons we see less expression in snRNA-seq data set that includes DMH and LH regions. We think that the scRNAseq dataset was more sensitive (capturing more genes per cell) and sufficient in providing the necessary information regarding the composition of the DMH. For this reason, we have excluded the snRNAseq dataset from our study.

9) CCK neurons in the DMH. As Imoto et al., 2021 describe, the DMH CCK neurons are specifically located in the compact portion of the DMH (cDMH). You can sort of see this too in the left DMH in Fig 5c as the diagonal band running from ventromedial to rostralateral. This will become even clearer in the slices more caudal to the one chosen to show in fig 6c. Could you find one more caudal that reflects this specific pattern of CCK being a marker for the cDMH better? Similarly, if you would zoom out a little in fig 6a, one would be able to appreciate this aspect. The fig 6a panels are rather large, so maybe a zoomed out image with the cDMH visible and an inset with the FOS detail could be a good option. Furthermore, you mention that "Cck-expressing neurons were predominantly in the

dDMH region," which does not correspond with the literature and the Allen brain atlas (see image 25 and 26 of the coronal CCK sections

<http://mouse.brainmap.org/experiment/siv?id=77869074&imageld=77809652&initImage=ish&coordSystem=pixel&x=6060m&y=5668&z=3>). Could you please address this? Do CCK neurons overlap with the DMH BRS3 population? Stimulation of BRS3 DMH neurons also increases EE (Pinol, 2018; PMID: 30349101).

We thank the reviewer for pointing at specific spatial expression of Cck in the DMH. We include new images showing Cck-expressing cells in cDMH and dDMH (Fig. 7). We observed that Cck positive cells in dDMH always preceded Cck positive cells in cDMH, meaning that Cck- positive cells in dDMH are present in the anterior portion of dDMH. In Fig. 7, we have carefully analyzed the presence of cFos^{Cck+} cells in these two subregions of DMH. Our DMH scRNA-seq data set analysis showed that Cck neurons do not overlap with Brs3 (Fig. 6i,j).

10) Since the implication of DMH CCK neurons in the MOB->PIR/OT/other?->DMH pathway is indirect ("Ttr clusters were abundant in Cck" and FOS staining in CCK in neurons), it could be worthwhile to increase evidence for this pathway or maybe somewhat attenuate the statements/conclusions that the DMH CCK neurons mediate the MOB activation/TMT induced VO₂ increase. Ideally one would a) inhibit/silence/kill the CCK neurons in an experiment Tbx21-Cre:hM3Dq experiment, or b) show with monosynaptic retrograde rabies tracing from CCK neurons and CRACM that the PIR/OT/other? neurons that project to DMH CCK neurons receive direct synaptic input from MOB neurons.

We now present new results where we silenced CCK neurons using viral delivery of tetanus toxin (Fig. 8 h-m). We have found that silencing DMH-CCK neurons reduced the amplitude of the energy expenditure response to TMT. Importantly, there remains a stress-induced heat production in DMH-CCK silenced mice which suggests that other neuronal chemotypes might contribute to this response (such as PVN-CRH neurons).

Quite interestingly, silencing DMH-CCK neurons did not alter the activity-induced burst following TMT suggesting that other neurons are involved in that behavioral response to stress. Rather, silencing DMH-CCK neurons increased feeding following TMT exposure as well as no odor, demonstrating an important role for these neurons in feeding. In the discussion, we acknowledge this partial phenotype "We also find that neuronal silencing of DMH^{CCK} neurons reduces TMT-induced gain in VO₂ (Fig. 8), but this reduced phenotype indicates that additional neurons might also contribute to this response."

We agree that CRACM would have been a good addition to our study, but because of technical limitations associated with mouse electrophysiological capacity at Cedars-Sinai we have not been able to use this technique.

Minor issues:

- This (p5) sounds a little garbled: "the activatory Gq DREADD receptor cholinergic receptor muscarinic 3". This is changed in the manuscript to "activatory Gq DREADD human M3 muscarinic acetylcholine receptors (hM3D)"
- Please specify if the Tbx21 Cre- controls are littermates of the Cre+ mice used in the experiment. Yes, we used Tbx21 Cre- littermates controls. We clarified this in the material and methods section, under the Animals paragraph, by stating, "In all experiments, appropriate littermate controls were used".
- Fig2o-q It is somewhat hard to measure Tbat without shaving the back of the mouse because the hair insulates. That is why the temperature is so low. We decided not to shave our mice following the practice

established by Crane et al. (<https://doi.org/10.1016/j.molmet.2014.04.007>).

- Sf2a, f4a and some other figs miss scale bar.

This has now been fixed.

- "declined after 3 hours and returned to baseline levels (Supp fig. 4d)" has to be S2d.

We changed to Supp fig. 2d

- Regarding the AAV-DIO-hM3Dq in Tbx21-Cre mice approach: "larger number of mitral cells are required". Could it be that the mitral cells that can drive EE are located in another part of the MOB?

Yes, this could be a plausible explanation for why we did not observe an increase in EE with this experimental approach. We now have changed the text as follows: "a larger number or different subpopulations of mitral cells"

- Sf2i: the increase in OCR in females is higher than in males, regardless of the genotype. What statistical test was used here? Also, is this an established phenomenon?

Two-way ANOVA and Three-way ANOVA were used for S fig 2i. Analysis showed the effect of sex ($p < 0.0001$) for both panels.

We have fixed the following errors suggested by the reviewer:

- "considered that Cre+ female's specific" is probably females'
- Fig 3h. Please add somewhere what the period of post CNO is?
- "The MeA has been heavily involved"; maybe "is" instead of "has been"?
- Fig4e. Since you are later focusing on the compact DMH, would it be possible to quantify the colocalization of Vglut2 and vgat with FOS in the subregions of the DMH (dorsal DMH, compact DMH, ventral DMH – and if you specify what exact region you refer to as caudal DMH)?
- "MOB of Ail14 (LSL-TdTomato reporter) mice" (p7) should be Ai14
- "in selected postsynaptic neuronal targets, thus allowing axonal tracing" should possibly be: "selectively in postsynaptic neuronal targets, allowing polysynaptic anterograde tracing"
- "identified 41 neuronal clusters (Fig. 5h)" should be 5g.
- "Our results suggest so far indicate that DMH neurons" is garbled

- Was the FOS expression in fig 6a and fig s7c quantified?

We now include Fig. 7 where cFos was quantified.

- I may have overlooked it, but is it mentioned anywhere if both uni and bilaterally expressing hMDdq mice were included in the analyses in the Slc1a3-Cre and CCK-Cre groups? I read that you injected the virus bilaterally, but that does not always translate in bilateral expression, irrespective of your brilliant stereotactic surgery skills.

We clarified this in Material and methods: "Verification of the Injection site is performed by IHC. Only animals with viral expression confirmed on both sites were used for experiments that required bilateral viral injection."

- The discussion says: "We therefore propose the neurocircuit MOB→VgatMeA→ DMH CCK" but this seems contradictory to some results in the manuscript. For instance you find that DMH CCK neurons are activated, not inhibited. Or are the MOB neurons also inhibitory?

Because we have not conducted CRACM to verify synaptic requirement, we have removed that sentence. We have included a summary Figure (Fig. 9) to explain our proposed model. In particular, we have included a putative intermediate neuron. We also include this statement in the discussion “We postulate that activating MeA^{Vgat} neurons might inhibit an unknown neuronal population which normally suppresses the activity of DMH^{CCK} neurons, therefore driving stress-induced thermogenesis (Fig. 9).”

- The last sentence of the discussion is a bit abrupt, since that is the first time in the manuscript that obesity is mentioned.

We changed it to “This neuronal circuit might be a potential therapeutic target for women’s stress and eating disorders”

Reviewer #3 (Remarks to the Author):

The manuscript by Jovanovic and colleagues presents intriguing evidence of a neural circuit involved in a female-specific response to a predator-related odor. Using a red fox predator scent, TMT, they show that female and male mice have divergent responses with respect to energy balance. Specifically, VO₂ consumption and activity are acutely elevated after exposure to TMT and then lowered during the following night. Feeding after an overnight fast is also suppressed by TMT in female mice. In contrast, VO₂ is acutely decreased in males and the effect on feeding after a fast is not as strong as in females.

The subsequent analyses are designed to identify brain regions, cells, and circuits that are involved, as measured by cFOS immunoreactivity, tracing, and transcriptomics. cFOS immunoreactivity is induced in the ARC and DMH, rather selectively by predator odor compared to other odors, but the effects are similar in males and females. FACS RNA-seq and snRNA-seq identify markers of populations within the DMH that are synaptically connected to TBX21+ olfactory bulb neurons. Finally, chemogenetic activation of TBX21-lineage and Cck-expressing neurons partially phenocopy the effect of TMT. Specifically, activating neurons of the TBX21+ lineage with a transgenic hM3Dq recapitulates some of the sex-specific phenotypes and activating Cck-expressing neurons using AAV-hM3Dq recapitulates some of the phenotypes in females, suggesting that activating a circuit that includes TBX21 OB neurons or Cck-expressing neurons of the DMH can partially recapitulate the effect of TMT. The authors also show that ovariectomy does not alter the effect activating neurons of the TBX21+ lineage, consistent with the conclusions that either the effects are not dependent on ovarian hormones or that chemogenetic neuronal activation can bypass any possible effects of ovarian hormones on the circuit.

I think that there are multiple major concerns that should be addressed prior to publication.

Major comments:

Ovariectomy did not block the effect of OB-Tbx21 activation on VO₂. This is interpreted to "indicate that this metabolic response is unlikely to be dependent on estrogens tuning of olfactory performance, but rather might be the consequence of sexually dimorphic neurocircuits". Although this is one possibility, it is also possible that neuron activation could bypass the need for or modulatory effect of gonads. For example, Mc4r-expressing neurons in the VMHvl regulate estrogen-dependent movement in females, but activating the neurons increases

movement in both sexes and in females without ovaries (PMID: 34646010). If the goal is to test the effect of gonads, it would be necessary to do the OVX experiment in the context of TMT odor presentation to ask if ovaries are required for the effects in females. Further, it would be important to include both sexes, with and without gonads, because it is possible that the effect is suppressed by testicular hormones in males. As presented, the OVX manipulation is difficult to interpret.

We agree with the reviewer that in order to fully rule out a role for gonads, gonadectomy assays in males should have been included. Yet, we have not performed that particular experiment due to limitations of time to address the many revisions required. In order to include this important point, we have modified our discussion as follows:

“Although we have not tested the presence of a stress-induced thermogenic response in males lacking gonads and therefore fully examine the role of sex hormones in our models, our findings suggest that this metabolic response is not influenced by the absence of estrogens, but rather might be the consequence of heightened stress integration in the female brain”.

The conclusion that "a neuronal population in the DMH [is] required for this response" is not supported by the data. Activating Cck+ DMH neurons can partially recapitulate the response in females but the phenotyping is more limited and the experiment does not include males. Even if the phenocopy were perfect, it would be more appropriate to conclude that the neuronal population in the DMH is sufficient for this response. If the goal is to test requirement, the authors would have to show that silencing or ablating Cck+ DMH neurons blocks the response to TMT.

Thank you for this pertinent suggestion. We have now included silencing experiments in Fig. 8 h-o viral delivery of tetanus toxin. These new data demonstrate that DMH-CCK neurons are partially required for the TMT-induced gain in energy expenditure.

Any claim of a sex difference requires sex (and interactions with sex) to be factors in an ANOVA or mixed model. The methods state that these statistical analyses were performed but I could not find the results of those tests.

We now have included all statistical analyses investigating interactions with sex in supplementary information (Supp. Tables S1-7).

Why would females, but not males, increase energy expenditure upon being exposed to predator odor? Is it related to escaping the odor by running away?

We provide some context in the discussion, as it is commonly observed in rodent models for females to be more responsive to certain forms of stress. Even though we do not provide supportive evidence to elaborate on the biological relevance of this phenotype, we speculate that this might be a strategy associated with female behaviors developed for offspring protection. Being more cautious could be evolutionarily beneficial to female mice. “ In rodent models, stressful stimuli tend to have more profound effects in females (55). Female rats have greater and more persistent CORT responses to stress (56). Rat parvocellular PVH innervation, measured by the

density of synaptophysin staining, is greater in females than in males (23). In addition, social isolation alters the intrinsic properties of CRH neurons in female, but not in male mice (57). In humans, there is also some evidence that stress-related disorders might be more prominent in women (22). “

To what extent is thermogenesis involved in the response to TMT? Thermogenesis is highlighted in the title but is only evaluated in the context of activating TBX21+ OB neurons. Is thermogenesis elevated by TMT? Is this a major effect of TMT or major output of the circuit? Why would TMT engage a thermogenic circuit?

Thank you for this comment. We now have added results showing that TMT induces a burst in BAT thermogenesis in female mice but not in males (Fig. 1i-j).

We have reformatted the introduction to include information regarding stress-hyperthermia:

“A key role of the stress response is the mobilization of the body’s energy stores to promote an adaptive response, such as fleeing a dangerous situation, however acute and chronic activation of stress share different outcomes on energy metabolism (5). Yet, the current understanding on the neurocircuits associated with the different metabolic consequences of acute and chronic stress is limited. Acute CRH release mobilizes energy stores, including increasing brown adipose tissue (BAT) thermogenesis, enhancing locomotion and suppressing feeding to ensure a rapid response favoring animal survival, in a process recognized as stress-induced hyperthermia (6–8).”

Minor Comments

The link to stress is highlighted in the title but is underdeveloped. There is a rise in cort levels but the response is not generalized to other stressful contexts. Perhaps the title and interpretations can be limited to the context of predator-related odor?

We have now changed the title to “A sex-specific thermogenic neurocircuit induced by predator smell recruiting Cholecystokinin neurons in the dorsomedial hypothalamus”.

Related to cort, the methods do not describe how the blood was collected for measurements in plasma. It would be important do describe how acute stress was avoided in the collection process to avoid inducing a stress response in all mice, including controls, which could mask an effect of TMT in males.

We updated the plasma corticosterone quantification section in Material and methods to state: “To minimize stress response in mice, animals were acclimated to the room for a few hours prior to treatment. The same experimenter performed handling during an experiment.”

The design of the chemogenetic experiments include saline injection but I cannot find the phenotyping of mice after saline injection. How does this compare to the effects of CNO in within-subject comparisons?

We have included saline injection data in Supp Fig.2. We found that Cre+ mice treated with saline did not exhibit changes in VO₂.

In the final figure, there are representative images of Cck transcript and cFOS immunoreactivity. It should be specified how many mice were imaged and the images should be quantified.

Thank you for recognizing this omission in our initial submission. In Fig 7., as recommended, we have included detailed quantification of cFos^{Cck+} cells. N=5/per group for panel b, and N=4/per group for panel d, were used for this quantification and we have provided this information in the figure legend.

REVIEWER COMMENTS

Reviewer #1 (Remarks to the Author):

The authors have performed numerous additional experiments, the results of which have largely strengthened the present study. Moreover, they have carefully edited the text according to the outcome of these novel experiments.

Reviewer #2 (Remarks to the Author):

The manuscript has significantly improved from the previous version. All my suggestions have been addressed with the exception that I have one more small suggestion for the scRNAseq dataset presentation. The new data are a nice addition.

Minor comments and suggestions:

- “Because LepR-DMH neurons are GABAergic ...” -- I think that ref 27 (Garfield et al 2016) showed that DMH->Arc neurons are GABAergic, but not necessarily all DMH LepR neurons. You also show later in fig 6 that lepR is expressed in the MC4R cluster.
- Fig. 6I,J: Thank you for adding more detail to the scRNAseq clustering dataset. Would it be possible to include all the transcripts that contribute to the names of the clusters in the violin plot? Eg, PDyn, Cartp, MC4R, etc are in the cluster names, but missing in the violin plot. It would make it much more interesting to read for researcher interested in DMH populations. I am aware of space issues, but maybe consider adding a “full” version to the supplement. Further, what is the reason that all cluster names have two gene names? Aren't there some with one or three? The methods section only says that the cluster were “analyzed by Seurat::FindMarkers to identify top genes.”
- Fig. 7. Thank you for clarifying the CCK neuron locations in the DMH. From the looks of it, without other landmarks, it appears that the images on the left in panels A and C are of the anterior DMH area, bregma -1.34 to -1.58, not necessarily from the dorsal part. I have not seen the dorsal DMH being drawn as a triangle (without the cDMH or vDMH) before either.

- Fig. 8j-k. The TeNT experiment is a nice addition to the manuscript and highlights the complex nature and multifactorial character of the fear odor induced physiology and behavior. “In TeNT animals, we failed to observe a burst in VO₂ upon swab introduction in the “no odor” condition which is present in eGFP controls” Perhaps if you take the 45 min timepoint you do. Is there a specific reason to take the 22.5 min time point for quantification? Do the TeNT mice have lower baseline VO₂? Taking one timepoint (t=0) is not a reliable quantification for baseline EE, physical activity or food intake. Maybe look at a period of several hours or an entire light period if you have the data. Similarly, was there a difference in baseline food intake between GFP and TeNT groups? And what about body weight? And physical activity?

Reviewer #3 (Remarks to the Author):

The authors have substantially improved the manuscript in this revision, particularly in the discussion of caveats and interpretations. The authors now explain the sex difference in the response to the odor and in the response to OBtbx21 activation in the context of published pre-clinical studies and clinical data suggesting differences in susceptibility to stress. They also acknowledge that there is no explanation for why a fox odor would be stressful (CORT, VO₂) to females but not males.

I particularly appreciate the addition of the silencing experiments to test for requirement. Figure 8, panels h-o shows a critical experiment for the claim that “DMHcck neurons are required for engaging both thermogenesis and suppressing feeding upon certain stressful olfactory cues” but the data and analyses are not quite convincing. Looking at 8j, it is unclear if there is a significant statistical effect of silencing the neurons on the response to odor. Supplementary table S7 doesn’t show a test for an interaction between odor and virus. And it is unclear why certain timepoints are selected for panels 8K and 8M.

Smaller points

The use of the word burst for an elevation in temperature is repeated and a bit odd. The elevation in temperature lasts 30-60 minutes? How do the authors define a burst in temperature (or thermogenesis)?

In the results section, the heading for the text that reports the results of Figure 4 says “in response to olfactory cues” but it should be “in response to OBtbx21 activation”. There should be a distinction between the response to activating this circuit and the response to the odor.

Dear Colleagues,

We thank you again for your comments and critiques of our study. As you will find, we have addressed the new points raised by the reviewers. We hope you will find our manuscript of interest for publication in Nature Communications.

Best regards,

Celine Riera

REVIEWER COMMENTS

Reviewer #1 (Remarks to the Author):

The authors have performed numerous additional experiments, the results of which have largely strengthened the present study. Moreover, they have carefully edited the text according to the outcome of these novel experiments.

We are grateful for your thoughtful comments and suggestions and for your help in improving our manuscript.

Reviewer #2 (Remarks to the Author):

The manuscript has significantly improved from the previous version. All my suggestions have been addressed with the exception that I have one more small suggestion for the scRNAseq dataset presentation. The new data are a nice addition.

We are thankful for this reviewer's helpful feedback and for suggestions.

Minor comments and suggestions:

- “Because LepR-DMH neurons are GABAergic ...” -- I think that ref 27 (Garfield et al 2016) showed that DMH->Arc neurons are GABAergic, but not necessarily all DMH LepR neurons. You also show later in fig 6 that lepR is expressed in the MC4R cluster.

We thank the reviewer for their thoughtful comment. We agree that not all DMH LepR neurons are GABAergic which is also validated by our DMH scRNA data. We modified the text as follows “Because a subset of GABAergic neurons which express LepR in the DMH are rapidly activated by food cue presentation^{27,35}, we examined their role in the response to both chemogenetic OBtbx21 and TMT stimuli.

- Fig. 6I,J: Thank you for adding more detail to the scRNAseq clustering dataset. Would it be possible to include all the transcripts that contribute to the names of the clusters in the violin plot?

Eg, PDyn, Cartp, MC4R, etc are in the cluster names, but missing in the violin plot. It would make it much more interesting to read for researcher interested in DMH populations. I am aware of space issues, but maybe consider adding a “full” version to the supplement. Further, what is the reason that all cluster names have two gene names? Aren't there some with one or three? The methods section only says that the cluster were “analyzed by Seurat::FindMarkers to identify top genes.”

We have included an extended graph in the Supplementary Data (Supp. Fig. 8) showing all the genes that generated the clusters' subcategories. Since there is no recognized standard for naming clusters in single-cell RNA sequencing data, we included a more detailed explanation of our approach in the material and method section of Single-cell RNA-sequencing of DMH: “For cluster nomenclature, neurons were first labeled as GABAergic or Glutamatergic. Two genes from the 25 most variable genes of each cluster were selected for generating the cluster's name. These 2 genes were selected based on their prior functional association with the DMH, whenever possible.”

- Fig. 7. Thank you for clarifying the CCK neuron locations in the DMH. From the looks of it, without other landmarks, it appears that the images on the left in panels A and C are of the anterior DMH area, bregma -1.34 to -1.58, not necessarily from the dorsal part. I have not seen the dorsal DMH being drawn as a triangle (without the cDMH or vDMH) before either.

We agree with the reviewer regarding the unusual triangular shape, and removed it from the right top panels of Figure 7a and 7c. Regarding the nomenclature of the subregions of the DMH, there exists some variability between Paxinos and Franklin's and Allen Brain Atlas, as this reviewer appreciates.

The reviewer is right concerning the top panels of Figure 7a and 7c being localized in the anterior DMH using Paxinos and Franklin's terminology. According to both Atlases, this region should contain the initial part of the dorsal DMH (also called anterior DMH in Allen Atlas which adds some confusion). To clarify, we have added the bregma reference in our figure.

We observed Cck-expressing neurons in dDMH (right top panel of Fig. 7a,c) approximately 100 microns anterior to cDMH shown on top left panel of Fig 7a,c. Based on this, we estimated that these neurons are located at earliest at approximately -1.7mm from bregma according to Paxinos and Franklin's Reference Mouse Brain Atlas, or around -1.55mm from bregma according to the Allen Mouse Brain Atlas. We have added the coronal reference to Bregma corresponding to the regions for the specific images that we selected. We added the statement “Cck-expressing neurons were found in the dorsal (dDMH) and compact (cDMH) regions, detectable from Bregma -1.7mm to Bregma -1.9mm.”

- Fig. 8j-k. The TeNT experiment is a nice addition to the manuscript and highlights the complex nature and multifactorial character of the fear odor induced physiology and behavior. “In TeNT animals, we failed to observe a burst in VO₂ upon swab introduction in the “no odor” condition which is present in eGFP controls” Perhaps if you take the 45 min timepoint you do. Is there a specific reason to take the 22.5 min time point for quantification? Do the Tent mice have lower baseline VO₂? Taking one timepoint (t=0) is not a reliable quantification for baseline EE, physical activity or food intake. Maybe look at a period of several hours or an entire light period if you have the data.

We understand the confusion regarding our quantification and have worked towards improving the clarity of the data presented here. The timepoint $t=22.5$ min was selected as it corresponds to the maximal response intensity upon introduction of the cotton swab and allows to compare all four groups at this time point. This timepoint corresponds to the maximal VO_2 measured upon swab introduction (Fig. 1c,d).

It is important to point out that data sampling with the Phenomaster system only occurs every 18 min. At $t=22.5$ min, all animals data have been collected for this time point. At $t=45$ min, we did not observe any difference in energy expenditure between groups (see figure 8j). Our sample size for the TeNT assay was smaller ($n=8$) than Figure 1 ($N=16$), therefore it is not too surprising that the effect measured were smaller than Figure 1 due to the variability of the olfactory test between “wild-type” individuals. Yet the sample size used in Fig. 8 h-o has measured statistically significant differences between groups using ANOVA.

We are arguing that DMH-CCK neurons are partially required for the TMT-dependent thermogenesis, as we still observe an increase in VO_2 between no odor/TeNT and TMT/TeNT, but the maximal VO_2 measured upon TMT introduction remains similar as no odor/eGFP (Fig. 8k) . This is stated in the Discussion: “We also find that neuronal silencing of DMH^{CCK} neurons reduces TMT-induced gain in VO_2 (Fig. 8j,k), but this reduced phenotype indicates that additional neurons might also contribute to this response.”

Similarly, was there a difference in baseline food intake between GFP and TeNT groups? And what about body weight? And physical activity?

There was no discrepancy in body weight between the groups, with TeNT weighing 20.43 ± 1.16 g and eGFP weighing 21.66 ± 0.87 g. Unfortunately, we did not collect data without swab introduction for food intake and activity.

We have now included the sentence : “Three weeks following surgeries, body weights were similar among TeNT and eGFP animals (body weights-TeNT = 20.43 ± 1.16 g; body weights-eGFP = 21.66 ± 0.87 g).”

Reviewer #3 (Remarks to the Author):

The authors have substantially improved the manuscript in this revision, particularly in the discussion of caveats and interpretations. The authors now explain the sex difference in the response to the odor and in the response to OBtbx21 activation in the context of published pre-clinical studies and clinical data suggesting differences in susceptibility to stress. They also acknowledge that there is no explanation for why a fox odor would be stressful (CORT, VO_2) to females but not males.

I particularly appreciate the addition of the silencing experiments to test for requirement. Figure 8, panels h-o shows a critical experiment for the claim that “DMHcck neurons are required for engaging both thermogenesis and suppressing feeding upon certain stressful olfactory cues” but the data and analyses are not quite convincing. Looking at 8j, it is unclear if there is a significant statistical effect of silencing the neurons on the response to odor. Supplementary table S7 doesn't

show a test for an interaction between odor and virus. And it is unclear why certain timepoints are selected for panels 8K and 8M.

We understand the confusion regarding our quantification and have worked towards improving the clarity of the data presented here. The timepoint $t=22.5$ min was selected as it corresponds to the maximal response intensity upon introduction of the cotton swab and allows to compare all four groups at this time point. This timepoint corresponds to the maximal VO_2 measured upon swab introduction (Fig. 1c,d).

It is important to point out that data sampling with the Phenomaster system only occurs every 18 min. At $t=22.5$ min, all animals data have been collected for this time point. Our sample size for the TeNT assay was smaller ($n=8$) than Figure 1 ($N=16$), therefore it is not too surprising that the effect measured were smaller than Figure 1 due to the variability of the olfactory test between “wild-type” individuals. Yet the sample size used in Fig. 8 h-o has measured statistically significant differences between groups using ANOVA. We initially only provided data for statistically significant factors and interactions. We have now included the additional tests for an interaction between odor and virus (3-way-ANOVA).

We are arguing that DMH-CCK neurons are partially required for the TMT-dependent thermogenesis, as we still observe an increase in VO_2 between no odor/TeNT and TMT/TeNT, but the maximal VO_2 measured upon TMT introduction remains similar as no odor/eGFP (Fig. 8k) . This is stated in the Discussion: “We also find that neuronal silencing of DMH^{CCK} neurons reduces TMT-induced gain in VO_2 (Fig. 8j,k), but this reduced phenotype indicates that additional neurons might also contribute to this response.”

Smaller points

The use of the word burst for an elevation in temperature is repeated and a bit odd. The elevation in temperature lasts 30-60 minutes? How do the authors define a burst in temperature (or thermogenesis)?

We agree with the reviewer and have changed “burst in temperature” by increased temperature or thermogenesis as appropriate.

In the results section, the heading for the text that reports the results of Figure 4 says “in response to olfactory cues” but it should be “in response to OBtbx21 activation”. There should be a distinction between the response to activating this circuit and the response to the odor.

We have changed the title to:

“Sexually dimorphic cFos expression in the MeA and DMH in response to chemogenetic stimulation of olfactory neurons.”

REVIEWERS' COMMENTS

Reviewer #3 (Remarks to the Author):

The authors have addressed all of my concerns. I only suggest softening the conclusion that DMH-CCK neurons are required for the phenotypes observed. As the authors state in the rebuttal, the silencing experiments suggest that neurons other than DMH-CCK neurons contribute to the response. But there are two places in the manuscript where the interpretation is too strong, such as: 1) The end of the abstract: "These results demonstrate that DMHCCK are required for the propagation of female stress-induced hyperthermia." 2) The end of the results section: "Taken together these results demonstrate that DMH-CCK neurons are required for engaging both thermogenesis and suppressing feeding upon certain stressful olfactory cues."

Dear Colleagues,

We thank you again for your comments and critiques of our study. As you will find, we have addressed the new points raised by the reviewers. We hope you will find our manuscript of interest for publication in Nature Communications.

Best regards,

Celine Riera

REVIEWER COMMENTS

Reviewer #3 (Remarks to the Author):

The authors have addressed all of my concerns. I only suggest softening the conclusion that DMH-CCK neurons are required for the phenotypes observed. As the authors state in the rebuttal, the silencing experiments suggest that neurons other than DMH-CCK neurons contribute to the response. But there are two places in the manuscript where the interpretation is too strong, such as: 1) The end of the abstract: "These results demonstrate that DMHCCK are required for the propagation of female stress-induced hyperthermia." 2) The end of the results section: "Taken together these results demonstrate that DMH-CCK neurons are required for engaging both thermogenesis and suppressing feeding upon certain stressful olfactory cues."

We have modified the manuscript to take into account the above suggestions. The abstract has been shortened to 150 words, and does not contain this sentence anymore.

The last sentence of the results section is now the following: "Taken together these results demonstrate that DMHCCK neurons participate in the generation of thermogenic and feeding responses driven by the detection of certain predator threats."